# MHC-I upregulation safeguards neoplastic T cells in the skin against NK cell-mediated eradication in mycosis fungoides

Yun-Tsan Chang [1], Pacôme Prompsy [1], Susanne Kimeswenger [2], Yi-Chien Tsai[1], Desislava Ignatova[3], Olesya Pavlova [1], Christoph Iselin[1], Lars E. French [4,5], Mitchell P. Levesque [3], François Kuonen [1], Malgorzata Bobrowicz [6], Patrick M. Brunner[7], Steve Pascolo [3], Wolfram Hoetzenecker [2,9] ✉ & Emmanuella Guenova [1,3,8,9] ✉

Cancer-associated immune dysfunction is a major challenge for effective therapies. The emergence of antibodies targeting tumor cell-surface antigens led to advancements in the treatment of hematopoietic malignancies, particularly blood cancers. Yet their impact is constrained against tumors of hematopoietic origin manifesting in the skin. In this study, we employ a clonality-supervised deep learning methodology to dissect key pathological features implicated in mycosis fungoides, the most common cutaneous T-cell lymphoma. Our investigations unveil the prominence of the IL-32β–major histocompatibility complex (MHC)-I axis as a critical determinant in tumor T-cell immune evasion within the skin microenvironment. In patients' skin, we find MHC-I to detrimentally impact the functionality of natural killer (NK) cells, diminishing antibody-dependent cellular cytotoxicity and promoting resistance of tumor skin T-cells to cell-surface targeting therapies. Through murine experiments in female mice, we demonstrate that disruption of the MHC-I interaction with NK cell inhibitory Ly49 receptors restores NK cell anti-tumor activity and targeted T-cell lymphoma elimination in vivo. These findings underscore the significance of attenuating the MHC-I-dependent immunosuppressive networks within skin tumors. Overall, our study introduces a strategy to reinvigorate NK cell-mediated anti-tumor responses to overcome treatment resistance to existing cell-surface targeted therapies for skin lymphoma.

Therapeutic approaches harnessing immune responses are hallmarks of modern oncology. Monoclonal antibodies (mAbs), specialized proteins designed to target tumor cell-surface antigens, have emerged as agents capable of eliminating malignant cells and inducing disease remission[1,2]. In addition to inducing tumor-cell apoptosis, therapeutic mAbs initiate immune effector signaling cascades that rely on the complement system and natural killer (NK) cell-mediated antibody-dependent cellular cytotoxicity (ADCC) as anti-cancer modes of action.

Mycosis fungoides (MF), a malignancy of skin-homing T cells, represents the most common form of cutaneous T-cell lymphoma (CTCL)[3]. Due to the high heterogeneity among malignant T cells in CTCL, both intra- and inter-individually[4–7], there is no single specific marker to unequivocally identify CTCL tumor cells. However, T-cell

receptors (TCRs), which are natural identifiers of T-cells, are routinely used to define patient-specific malignant T-cell clones[8-11].

Progressive impairment of cell-mediated immunity is a hallmark of cancer, and in patients with CTCL, inadequate immune response has been reported[12]. Tumorous T cells play a significant role in suppressing the patient's immune system, thereby blunting the response of other immune cells that would otherwise mount an anti-tumor defense. This impairment of cellular immunity may negatively affect all therapeutic approaches that rely on a functional immune system. Modern therapeutic strategies based on mAbs that target specific T-cell subsets or multiple blood cell lineages are now established for the treatment of CTCL[13,14]. For instance, anti-CD52-mAb targets CD52, a pan-lymphocytic transmembrane glycoprotein expressed on mature T and B lymphocytes and a subset of monocytes and dendritic cells, while anti-CCR4-mAb targets CCR4$^+$ skin-homing lymphocytes. These therapeutic mAbs are highly effective at treating blood tumoral disease in patients with Sézary syndrome (SS; also known as leukemic CTCL, L-CTCL), with anti-CCR4-mAb treatment showing increased efficacy in patients with high blood tumor burden[15]. Interestingly, however, both mAbs have been shown to be less effective in treating skin lesions in patients with MF[16-20].

Immune checkpoint blockade represents a promising approach to restore the anti-tumor responses of endogenous immune cells. NK-cell inhibition-blocking mAbs acting as NK-immune checkpoint inhibitors show great potential when combined with NK-mediated anti-cancer therapy[21]. The anti-KIR2DL1/2/3-mAb-IgG4 targets inhibitory KIR2DL1/2/3 receptors on NK cells and blocks their interaction with human leukocyte antigen (HLA)-C (classical MHC-I), thereby enhancing NK-mediated anti-tumor activity[22,23]. Likewise, the anti-NKG2A-mAb-IgG4 targets inhibitory NKG2A receptors on NK cells and disrupts their interaction with HLA-E (nonclassical MHC-I), strengthening NK-mediated anti-tumor responses[24,25]. Both anti-KIR2DL1/2/3-mAb-IgG4[23] and anti-NKG2A-mAb-IgG4[26] have demonstrated the enhancement of NK-mediated ADCC and anti-tumor immunity.

In this study, we functionally characterize the tumor intrinsic factors responsible for impaired cellular immunity and resistance to skin therapies. We perform single-cell sequencing of skin T cells from patients with MF patients, mathematically reconstruct the TCRs of individual T cells, and categorize them into clonal and non-clonal groups based on TCR similarity. Using a clonality-supervised deep learning methodology, we develop a neural network logistic regression (NN-log-reg) machine-learning (ML) method to predict key tumor-related genes. Functional analysis of these genes reveals a high expression of MHC-I, potentiated by interleukin (IL) 32β, on skin tumor T cells as a mechanism rendering targeted therapy ineffective in MF. We confirm the relevance of these findings both ex vivo and in vivo.

## Results

### A clonality-guided deep learning approach to identify genes with predictive value for cancer in skin tumor T cells

Within MF lesions, the population of skin T cells consists of both malignant T cells and a variable count of benign bystander skin T cells[27,28]. For a thorough analysis of the skin tumor-specific transcriptome, we conducted single-cell RNA sequencing (scRNA-seq) on total T cells isolated from MF skin lesions and blood of MF patients with blood involvement, as well as T cells from healthy individuals, all pre-enriched using fluorescence-activated cell sorting (FACS) for live CD45$^+$/CD3$^+$ T cells. Leveraging TCR sequences as inherent clonality markers, we implemented a computational approach to segregate T cells from MF skin lesions into distinct clonal malignant and non-clonal, heterogeneous bystander T-cell populations. This was achieved by mathematically reconstructing complete, paired α and β sequences of the TCR for each individual cell and subsequent use of these sequences as labels for clonality-guided deep learning approach. This served as the foundation for training ML methods to detect genes with

predictive value for cancer in skin tumor T cells. Furthermore, it allowed for subsequent single-cell RNA transcriptomic analysis and predictions of disease-specific clinical outcomes at the level of individual T cells (Fig. 1a).

We conducted scRNA-seq using the Fluidigm C1 platform, known for its superior sensitivity and high-quality results[29], particularly apt for TCR reconstruction and concurrent transcriptome analysis of individual T cells. However, a major constraint of the Fluidigm C1 platform is its capacity to capture and process a maximum of 96 individual cells per sample. Consequently, with this method, we were able to profile a cumulative count of no more than 1174 high-quality T cells extracted from 17 tumor skin lesions belonging to 14 patients with MF. Additionally, we collected 573 blood T cells from three out of the 14 patients with MF who exhibited blood involvement and 192 T cells from four healthy individuals. The clinical characteristics of the patients are provided in Supplementary Table 1.

The cell capture rate reached 87.2%, with a corresponding qualification rate of 90.7% among the captured cells. The TCR reconstruction rates were 48.1% for the α chain and 71.8% for the β chain (Supplementary Table 2). T cells that successfully passed the FastQC control exhibited a low mitochondrial content and a high number of detected genes (Supplementary Fig. 1a). Healthy T cells demonstrated the expected high level of TCR diversity in their V-(D)-J combinations. In contrast, skin T cells from patients with MF displayed dominance in at least one V-(D)-J combination, signifying the presence of malignant clones. "ImMunoGeneTics" (IMGT), serves as an established resource offering a standardized nomenclature for genes and alleles, encompassing diverse variable immune structures, such as TCRs. According to the IMGT gene nomenclature, each human variable α-/β- chain of the TCR protein is precisely designated by a corresponding TCRα/β V gene (TRAV-/TRBV-). To visually represent the protein-level aspects of the computationally defined dominant TCRβ sequence within the primary malignant T-cell clone, we relied on the IMGT nomenclature to specify the exact Vβ protein segments on the surface of T cells (Supplementary Table 3). This methodology allowed us to conduct Vβ clonality assessment using flow cytometry for each patient sample. Among the 14 patients with MF, the corresponding TCR Vβ protein chain aligned with the predominant V gene was readily visualized in 10 cases (Supplementary Fig. 1b). In the case of the remaining 3 out of 4 patients, identification of the malignant clones relied solely on mathematical TCR reconstruction, underscoring the enhanced sensitivity of the sequencing approach for detecting T-cell clonality (Supplementary Table 3). Mathematical TCR reconstruction achieved a detection rate of 92.9% (13 out of 14) for malignant clones (Fig. 1b), outperforming the recognition rate of 71.4% (10 out of 14) obtained through flow-cytometric TCR Vβ chain detection. A single case (7.1%, 1 out of 14) did not yield detection of a malignant clone using either method. Consequently, this case has been excluded from subsequent analyses.

Subsequently, we established clonality criteria (see Methods), allowing us to further categorize the T cells into two populations linked to the main clone, namely "main-clone" and "related-to-main-clone", as well as two nonclonal bystander populations, referred to as "bystander groups" and "single bystanders" (Fig. 1c; Supplementary Fig. 2). After excluding CD8-expressing TCR-reconstructed T cells, the classification yielded 63.8% of all CD4$^+$ T cells in the "main-clone", 17.1% in the "related-to-main-clone", and 6.1% and 13.1% in the "bystander groups" and "single bystanders", respectively.

For a comprehensive single T-cell RNA transcriptomic analysis of skin T cells, we implemented a NN-log-reg model, an adaptive logistic regression model for which the model weights were set with a neural network, to identify genes with predictive potential for cancer in skin MF tumor T cells (Fig. 1d). Given that T-cell clonality was primarily inferred from TCR genes (clonality-guided approach), we omitted all TCR genes from the analysis. First, we conducted a comparative analysis of our NN-log-reg method with other machine learning

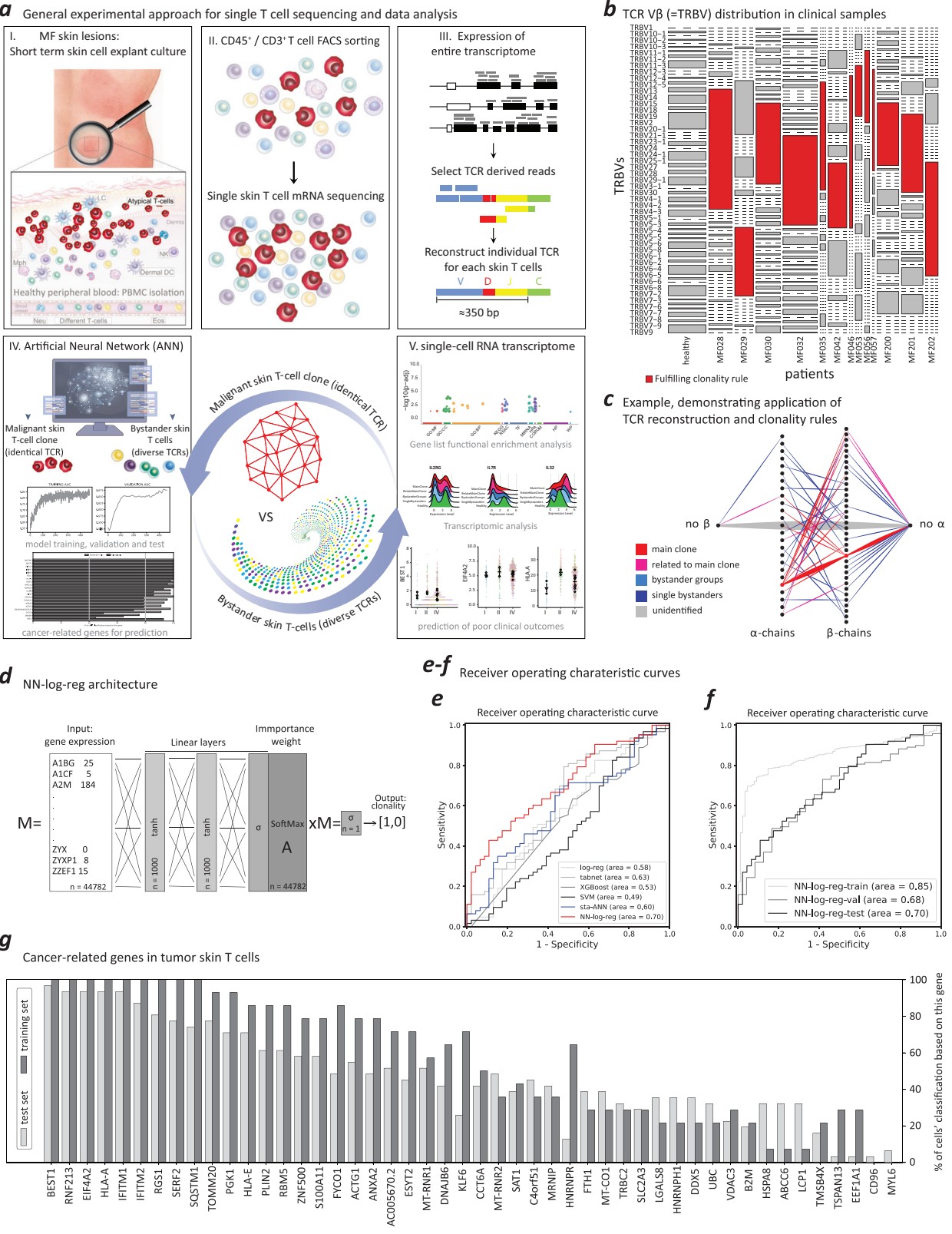

*a* General experimental approach for single T cell sequencing and data analysis

*b* TCR Vβ (=TRBV) distribution in clinical samples

*c* Example, demonstrating application of TCR reconstruction and clonality rules

*d* NN-log-reg architecture

*e-f* Receiver operating charateristic curves

*g* Cancer-related genes in tumor skin T cells

approaches: a standard artificial neural network[30] (sta-ANN), logistic regression[31] (log-reg), support vector machines[31] (SVM), neural network for interpretable classification of tabular data (tabnet[32]) and gradient-boosted tree model (XGBoost[33]). By training these models on skin cells from all patients, we observed that most models successfully discerned the category of skin CD4+ T cells (Supplementary Fig. 3a). The various machine learning methods displayed distinct learning behaviors (Supplementary Fig. 3b). However, our NN-log-reg method demonstrated superior performance compared to all other tested approaches (Fig. 1e). It exhibited strong generalization and, notably, maintained nearly equivalent performance on both the validation and test sets (Fig. 1f). When assessing the robustness of the NN-log-reg method, we identified a significant positive correlation between the network's prediction error and its precision (Supplementary Fig. 3c).

**Fig. 1 | A clonality supervised deep-learning approach to identify cancer-predicting genes in skin tumor T cells. a** General experimental approach for single T-cell sequencing and data analysis. Generation of scRNA-seq data from MF skin lesions and healthy blood samples, FACS sorting and computational segregation of total blood and skin T-cell populations using natural clonality identifiers, the T-cell receptor sequences, as labels in a supervised approach. The artificial neural network and single T-cell RNA transcriptomic analysis were subsequently based on computational segregation of clonal and nonclonal skin T-cell populations. The co-author O.P. creates the schematic representations. TCR reconstruction and clonality rules. **b** The distribution of T-cell receptor β variable genes, as determined by scRNA-seq data from clinical samples, demonstrates the heterogeneity of MF ($N = 13$ patients). Malignant clones in each sample are labeled as red; the healthy sample (collected from $N = 4$ individuals) was used as a control. **c**, Clonogram illustrating the classification of two clone-related populations, 'main-clone' and 'related-to-main-clone', and two nonclonal bystander populations,

'bystander groups' and 'single bystanders'. The detailed illustrations of the clonograms refer to Supplementary Fig.2. ANN and single T-cell RNA transcriptomic analysis of skin T cells ($N = 825$ cells from 13 patients). **d** Architecture of the NN-log-reg. M represents the original expression value of each gene in the initial list as detected in the single T-cell sequencing, while A represents the importance for prediction value of each gene in the important weights. **e**, The receiver operating characteristic curve (ROC) of different ML methods on the skin test set, composed of cells from donors previously unseen by the models. **f** The ROC of the NN-log-reg method on skin in training, validation and test sets. **g** The 48 overlapping, cancer-related genes identified by the training and test sets of NN-log-reg method in tumor skin T cells. The x axis indicates the percentage of cells for which the specific gene (on the y axis) is among these top 0.5% in the 'feature importance' list. For instance, a value of 100% for "gene A" means that "gene A" is among the top 0.5% of 'important genes' for 100% of correctly classified cells. Source data are provided as a Source Data file.

We configured our NN-log-reg architecture to predict whether each T cell derived from a patient with MF was either malignant or a bystander, relying exclusively on non-TCR-related genes within the transcript subset. Beyond ascertaining the distinction between malignant and benign cell types, we additionally trained the NN-log-reg to recognize all transcripts factored into its considerations for precise predictions (Supplementary Fig. 3d). These predictions were exclusively based on cells that were accurately classified. Among the significant clonal genes identified, 79 were discerned by the training set, and 50 were identified by the test set, with 48 genes overlapping between both sets (Fig. 1g).

### MHC-I is significantly upregulated on malignant T cells from MF skin lesions

To gain mechanistic insights into the list of 48 overlapping and important genes with predictive value for cancer, we performed a gene set enrichment analysis. Two key pathways linked to tumor T cells in MF were identified: (1) the MHC class I protein complex; and (2) negative regulation of NK cell-mediated immunity (Fig. 2a).

For all 14 patients with MF, clinical characteristics and disease stage were assessed at baseline (sample acquisition) and 5 years later or at death, whichever came first (Supplementary Table 1). All patients were diagnosed with early-stage disease (stage IA, IB or IIA) at the time of sample collection. This prompted an exploration into transcriptome dynamics in individual tumor cells. Single-cell transcriptomic analysis depicted the disease progression of tumoral skin T cells based on the real-time disease stage that patients with MF reached either at death or within a 5-year follow-up period. Leveraging this insight, we evaluated the gene expression of some of the most important genes (identified via the NN-log-reg method [Fig. 1g]) on their real-time disease stage and observed notable escalation in the expression of *MHC-I*, *IFITM1* and *IFITM2* as disease progressed (stage IIB-IV) compared to early stage (stage I-IIA) (Fig. 2b).

Furthermore, this current analysis extended our recently published single-cell RNA dataset [GSE173205][34], encompassing patch (early stage) and plaque/tumor (late stage) skin lesions from three patients with MF. This dataset, derived from 10X single-cell RNA sequencing with TCR information, offered a substantial number of sequenced single cells for identifying each patient's malignant clone. By comparing differentially expressed genes (DEGs) of malignant clonal skin tumor T cells between patch and plaque/tumor lesions of the three patients, we pinpointed genes that exhibited an increase with disease progression. This alternative approach to scRNA-seq using the C1 Fluidigm platform also highlighted a statistically significant increase in the expression of three classical MHC-I genes—*HLA-A*, *HLA-B*, and *HLA-C*—in tumor T cells from plaque/tumor late-stage skin lesions (Supplementary Fig. 4a).

To validate these findings functionally, we assessed the protein expression levels of HLA-A, HLA-B, and HLA-C, the non-classical MHC-I

(HLA-E), as well as IFITM1 and IFITM2 in T cells derived from both healthy skin and MF skin lesions using Western blot analysis. All three HLA-A, HLA-B, and HLA-C exhibited a highly significant increase in expression on T cells from MF skin lesions when compared to T cells from healthy skin (Fig. 2c). While HLA-E also displayed a significant increase on T cells from MF skin lesions, its expression remained lower than that of the three classical MHC-I proteins (Fig. 2c). Given the minimal detectable expression of the remaining two genes (IFITM1 and IFITM2) (Supplementary Fig. 4b), they were excluded from subsequent analysis.

Consequently, we assessed classical and non-classical MHC-I expression on bystander and malignant T cells derived from both skin and blood of three patients with late-stage MF and blood involvement (L-CTCL) via flow cytometry. Interestingly, the mean fluorescence intensity (MFI) of classical MHC-I (HLA-ABC) displayed a noteworthy increase on skin malignant T cells (Fig. 2f). However, the expression of non-classical MHC-I (HLA-E) remained low, with no statistically significant difference observed between samples (Fig. 2d; Supplementary Fig. 4c). This observation directed our focus toward classical MHC-I (HLA-ABC).

We used flow cytometric analysis to assess classical MHC-I expression on viable bystander and malignant T cells derived from both skin and blood of patients with MF and L-CTCL, as well as benign T cells from the skin and blood of healthy individuals. The proportion of classical MHC-I^high T cells was higher in the MF malignant T-cell population compared to all other studied T-cell populations (Fig. 2e). The mean fluorescence intensity (MFI) of classical MHC-I was most pronounced exclusively in MF skin malignant T cells (Fig. 2f). Furthermore, we performed a comparative analysis of different skin tumor types, such as basal cell carcinoma (BCC), squamous cell carcinoma (SCC), cutaneous B-cell lymphoma (CBCL) and melanoma (Fig. 2g). This confirmed our computational prediction for clear and statistically significant upregulation of inhibitory classical MHC-I protein specifically on malignant T cells from MF skin and allowed us to postulate a skin- and T-cell lymphoma-specific finding.

A more detailed intra-individual pair-wise comparison proved that MHC-I is upregulated in malignant T cells from MF skin lesions in each individual patient (Supplementary Fig. 4d). Moreover, CD45 expression remained comparable between bystander and malignant T cells of patients with L-CTCL and MF (Supplementary Fig. 4e); however, we observed a tendency toward decreased CD3 expression on malignant T cells (Supplementary Fig. 4f), as previously reported[35].

The classical MHC-I molecule is not only important for antigen presentation, but also as an inhibitory ligand for NK cells, key effectors in tumor cell-surface antigen-targeted therapy (Supplementary Fig. 4g). Indeed, our gene set enrichment analysis uncovered both MHC-I and negative regulation of NK cell-mediated immunity as key pathways linked to tumor T cells (Fig. 2a). Hence, we hypothesized that classical MHC-I, highly expressed on tumor skin T cells, negatively regulates NK cell-mediated immunity and ADCC activity in MF skin.

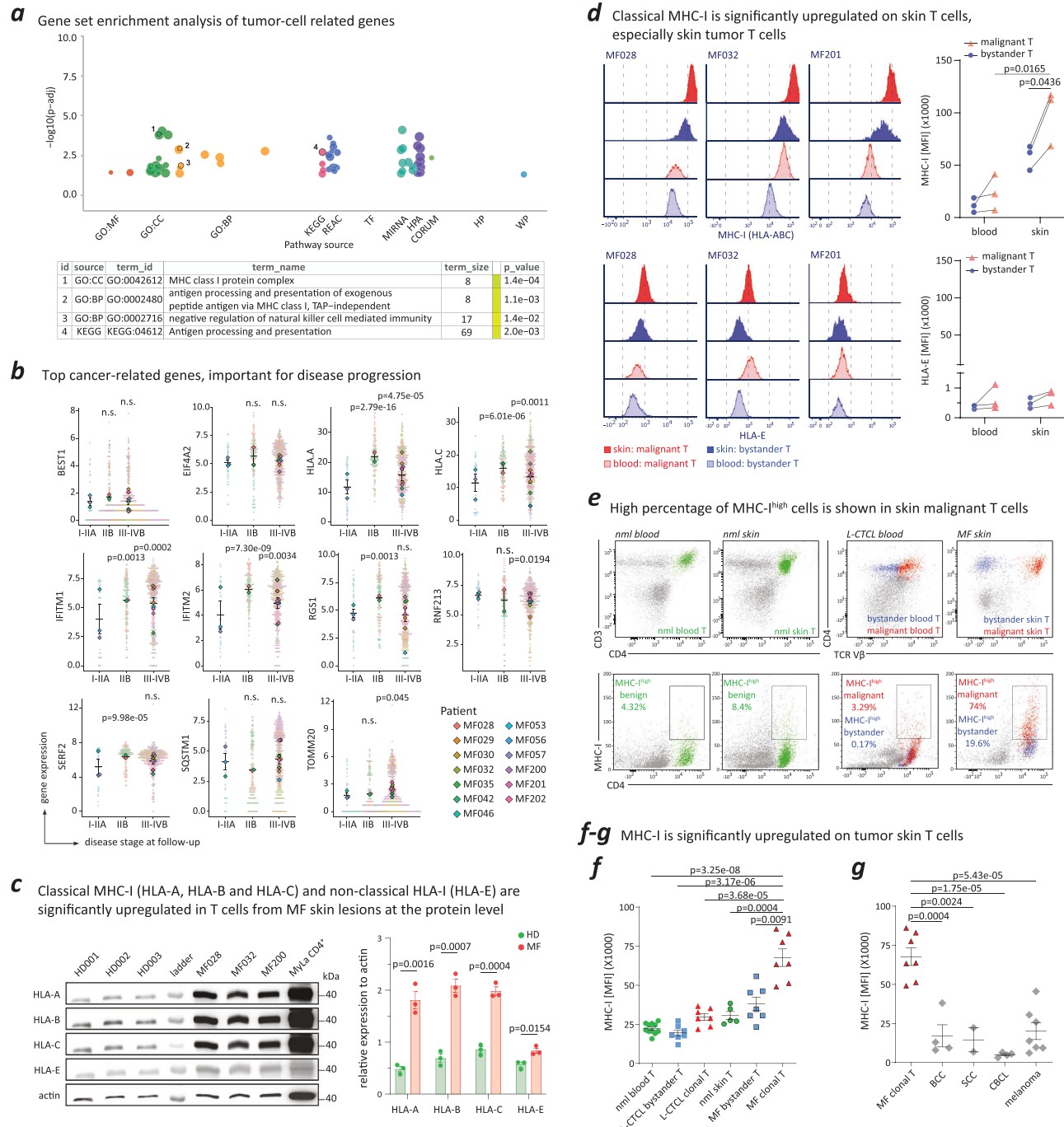

**a** Gene set enrichment analysis of tumor-cell related genes

| id | source | term_id | term_name | term_size | p_value |
|---|---|---|---|---|---|
| 1 | GO:CC | GO:0042612 | MHC class I protein complex | 8 | 1.4e-04 |
| 2 | GO:BP | GO:0002480 | antigen processing and presentation of exogenous peptide antigen via MHC class I, TAP-independent | 8 | 1.1e-03 |
| 3 | GO:BP | GO:0002716 | negative regulation of natural killer cell mediated immunity | 17 | 1.4e-02 |
| 4 | KEGG | KEGG:04612 | Antigen processing and presentation | 69 | 2.0e-03 |

**b** Top cancer-related genes, important for disease progression

**c** Classical MHC-I (HLA-A, HLA-B and HLA-C) and non-classical HLA-I (HLA-E) are significantly upregulated in T cells from MF skin lesions at the protein level

**d** Classical MHC-I is significantly upregulated on skin T cells, especially skin tumor T cells

**e** High percentage of MHC-I^high cells is shown in skin malignant T cells

**f-g** MHC-I is significantly upregulated on tumor skin T cells

## Classical MHC-I negatively impacts NK cell-mediated ADCC activity against malignant T cells in the skin

Therapies that rely on ADCC are effective against hematopoietic tumors, particularly B-cell malignancies. Anti-CD20-mAb, which targets the B-cell surface marker CD20, provides a documented clinical response against skin-limited B-cell tumors[36,37], and this activity of depleting B cells in the skin is paralleled by depletion of B lymphocytes in the blood (Fig. 3a).

Conversely, T-cell tumors in the skin do not respond well to therapies targeted against tumor cell-surface antigens. Skin-limited MF has only a moderate response to both anti-CD52-mAb and anti-CCR4-mAb, even though the targeted CD52⁺/CCR4⁺ lymphocyte population is successfully depleted in the blood (Fig. 3b)[16,19,38,39]. In a rare clinical case of advanced MF, B-cell–specific CD20 was aberrantly expressed on tumor skin T cells (Supplementary Fig. 5a, b); this prompted the

initiation of anti-CD20-mAb therapy as a personalized treatment. While this treatment depleted B lymphocytes in the blood, skin symptoms worsened after treatment initiation, despite apparent reduced CD20 expression on skin tumor cells (Fig. 3c; Supplementary Fig. 5c).

NK cells play a crucial role as effector cells in ADCC[40], and their presence within the skin is pivotal for the efficacy of treatment. For insights into the proportion of skin NK cells across healthy skin and various skin disorders, we analyzed our previously published and now publicly available single-cell RNA datasets, encompassing MF [GSE173205[34] and GSE165623[41]], atopic dermatitis [GSE222840[42]], BCC [GSE181907[43]] and CBCL [GSE173820[44]]. We defined NK cells using the gene signature comprising *NKG7, KLRB1, KLRC1, KLRD1, KLRK1, CD7, GZMB, GNLY, NCAM1, GZMH, CCL4, IFNG, CCL4L2, FCGR3B*, and *FCGR3A* (Supplementary Fig. 6a). The average percentage of skin NK cells within

**Fig. 2 | MHC-I is significantly upregulated on malignant T cells from MF skin lesions. a** Gene set enrichment analysis of overlapping, important tumor (clonal) cell-related genes ($N = 48$) discovered two important pathways, the MHC class I protein complex and negative regulation of NK cell-mediated immunity. **b** Gene expression of some of the most important genes (identified via the NN-log-reg method [Fig. 1g]) on their real-time disease stage at follow-up indicated notable escalation in the expression of MHC-I, IFITM1 and IFITM2 as disease progressed. Each dot represented normalized expression of the given marker in a single cell. The larger solid points represented mean normalized expression per patient sample. $N = 57$ cells from 3 patients were used in stage I-IIA; $N = 321$ cells from 2 patients were used in stage IIB; $N = 814$ cells from 8 patients were used in stage III-IVB. Data were presented as mean values +/− SEM of the larger solid points in each stage. For HLA-A and HLA-C, the normalized expression of all transcripts was summed. The $p$ values were calculated using unpaired, two-tailed student's $t$ test to compare normalized expression at the single-cell level taking the early stage (I-IIA) as reference. n.s., not significant. **c** Western blot analysis revealing that protein expressions of three classical MHC-I molecules (HLA-A, HLA-B, and HLA-C) were significantly higher on T cells from MF skin lesions (N = 3) compared to healthy skin lesions ($N = 3$). Conversely, the expression of the non-classical MHC-I molecule (HLA-E) was relatively low on T cells from MF skin lesions. Data were presented as mean values +/− SEM. The $p$ values were calculated using unpaired, two-tailed student's $t$ test. **d** Classical MHC-I (HLA-ABC) expression markedly increased on malignant skin T cells ($N = 3$) compared to bystander T cells in the skin ($N = 3$) and malignant T cells in the blood ($N = 3$). Non-classical MHC-I (HLA-E) expression remained low and showed no statistical difference between these samples ($N = 3$). The $p$ values were calculated using paired, two-tailed student's $t$ test. Gating strategy is shown in supplementary Fig. 10. **e** A higher percentage of MHC-I$^{high}$ T cells was present in malignant T-cell populations from MF skin lesions versus other T-cell populations (representative example of $N = 7$). **f** Strong expression of MHC-I on malignant T cells from skin lesions of patients with MF ($N = 7$) compared with bystander T cells from MF skin lesions ($N = 7$) or benign T cells from the skin of healthy individuals ($N = 5$). The expression of MHC-I on malignant T cells in MF skin lesions ($N = 7$) was also stronger than on malignant and bystander T cells from the blood of patients with L-CTCL ($N = 7$) and benign blood T cells of healthy individuals ($N = 11$). Data were presented as mean values +/− SEM. The $p$ values were calculated using unpaired, two-tailed student's $t$ test. Gating strategy is shown in supplementary Fig. 10. **g** In addition, the MFI of MHC-I in malignant T cells from MF skin lesions showed significantly greater intensity than in other types of skin tumors, such as basal cell carcinoma (BCC, $N = 4$), squamous cell carcinoma (SCC, $N = 2$), cutaneous B-cell lymphoma (CBCL, $N = 4$) and melanoma ($N = 7$). Data were presented as mean values +/− SEM. The $p$ values were calculated using unpaired, two-tailed student's $t$ test. Source data are provided as a Source Data file.

the immune-cell population was ~2%, with no significant variance observed between diseases (Supplementary Fig. 6b). Next, employing multiparameter immunofluorescence staining, we visualized the NK cells in both healthy and MF skin lesions (Fig. 3d). In the studied lesions, the average percentage of NK cells was higher in the skin from patients with MF compared to healthy individuals (Fig. 3e). Histologically, CD56$^+$ NK cells were readily distinguishable amid tumor T cells in MF skin lesions (Supplementary Fig. 6c). For in vitro experiments, we isolated viable total skin single cell suspensions and specifically sorted the CD45$^+$ lymphocyte population from MF skin lesions. We then segregated malignant and bystander skin T cells based on patient-specific TCR Vβ expression and skin NK cells using CD3$^-$ and CD56$^+$ expression (Fig. 3f).

Functional analysis showed, while autologous NK cells effectively initiated an ADCC response against L-CTCL blood tumor T cells, their ADCC potency against MF skin tumor T cells was compromised (Fig. 3g). More specifically, autologous skin NK cells elicited a potent ADCC response against both healthy and MF bystander T cells, yet this response was suppressed against MF malignant T cells (Fig. 3h). Microscopically, ineffective ADCC activity resulted in a lack of visible activity clusters against MF skin T cells, irrespective of the concentration of the therapeutic antibody. At the same time, the presence of activity clusters visibly and concentration-dependently increased as a marker for effective NK cell-mediated ADCC against malignant T cells from L-CTCL blood (Supplementary Fig. 7a).

Drawing from these findings, we postulated a significant role in this mechanism for the prominently expressed classical MHC-I on tumor skin T cells. Our hypothesis gained additional support as we excluded alternative resistance mechanisms. For example, we ruled out trogocytosis, a process that can hinder ADCC by cleaving the targeted antigen of tumor cells (Supplementary Fig. 7b, c). Furthermore, we evaluated the presentation and effectiveness of the complement system activation induced by targeted therapy. In patients with L-CTCL and MF, serum C1q levels and the functionality of complement-dependent cytotoxicity (CDC) did not significantly differ from those observed in healthy individuals (Supplementary Fig. 7d, e). Therefore, and building upon our earlier findings, we focused on investigating whether blocking classical MHC-I could restore NK cell functionality against malignant T cells from MF skin lesions.

### Classical MHC-I blockade restores NK cell-mediated ADCC activity against malignant T cells in MF skin lesions

In vitro, the inhibition of classical MHC-I effectively reinstated ADCC activity in autologous NK cells against MF malignant T cells (Fig. 4a).

This confirmed the functional relevance of our findings and validated our initial hypothesis.

To further decipher this phenomenon, we used carboxyfluorescein succinimidyl ester (CFSE) to label NK cells, which emitted green fluorescence, while benign and MF malignant T cells were marked with Far Red dye (red fluorescence). We observed discernible clusters of activity emerged within the autologous NK cell group, signifying effective ADCC responses. These clusters revealed that MF malignant T cells, opsonized by a cell surface-targeted antibody, were encircled by autologous NK cells during classical MHC-I blockade, facilitated by either anti-classical-MHC-I IgG or anti-classical-MHC-I F(ab')2 (Fig. 4b). Furthermore, the corresponding ADCC activity measurements for each group demonstrated that the visible clusters accounted for the majority of the ADCC response (Fig. 4c).

These results unequivocally validated the restoration of NK cell-mediated ADCC activity against MF malignant skin T cells through classical MHC-I blockade. However, due to the essential role of classical MHC-I in antigen presentation, directly inhibiting classical MHC-I on skin tumor T cells is not viable in practical real-world scenarios. To overcome this challenge, a promising solution lies in targeting inhibitory killer cell immunoglobulin-like receptors (KIRs) on NK cells. This approach holds significant potential to disrupt the interaction between classical MHC-I and KIR, presenting an option for further exploration as a potential treatment strategy for patients.

We used a human KIR-blocking IgG4 (lirilumab) to specifically target inhibitory KIR2DL1/L2/L3, which interacts with HLA-C of classical MHC-I. As anticipated, inhibitory KIR2DL1/L2/L3 blockade resulted in a significant restoration of NK cell-mediated ADCC activity against MF malignant T cells, irrespective of the specific cell-surface antigen being targeted. This effect was consistently observed across two different cell-surface targeted antibodies relevant for CTCL (anti-CD52-mAb and anti-CCR4-mAb), which was validated through two distinct experimental approaches: flow cytometry (Fig. 4d and Fig. 4e) and the lactate dehydrogenase (LDH) release assay (Fig. 4f).

### Inhibiting the interaction between classical MHC-I and Ly49-C/I enhances NK cell-mediated anti-T-cell lymphoma activity in mice skin in vitro and in vivo

Subsequently, we employed murine model systems to investigate the hypothesis that targeting the blockade of classical MHC-I and its inhibitory receptor on NK cells holds therapeutic potential for CTCL treatment. Ly49, analogous to human KIR, binds to H-2Kb

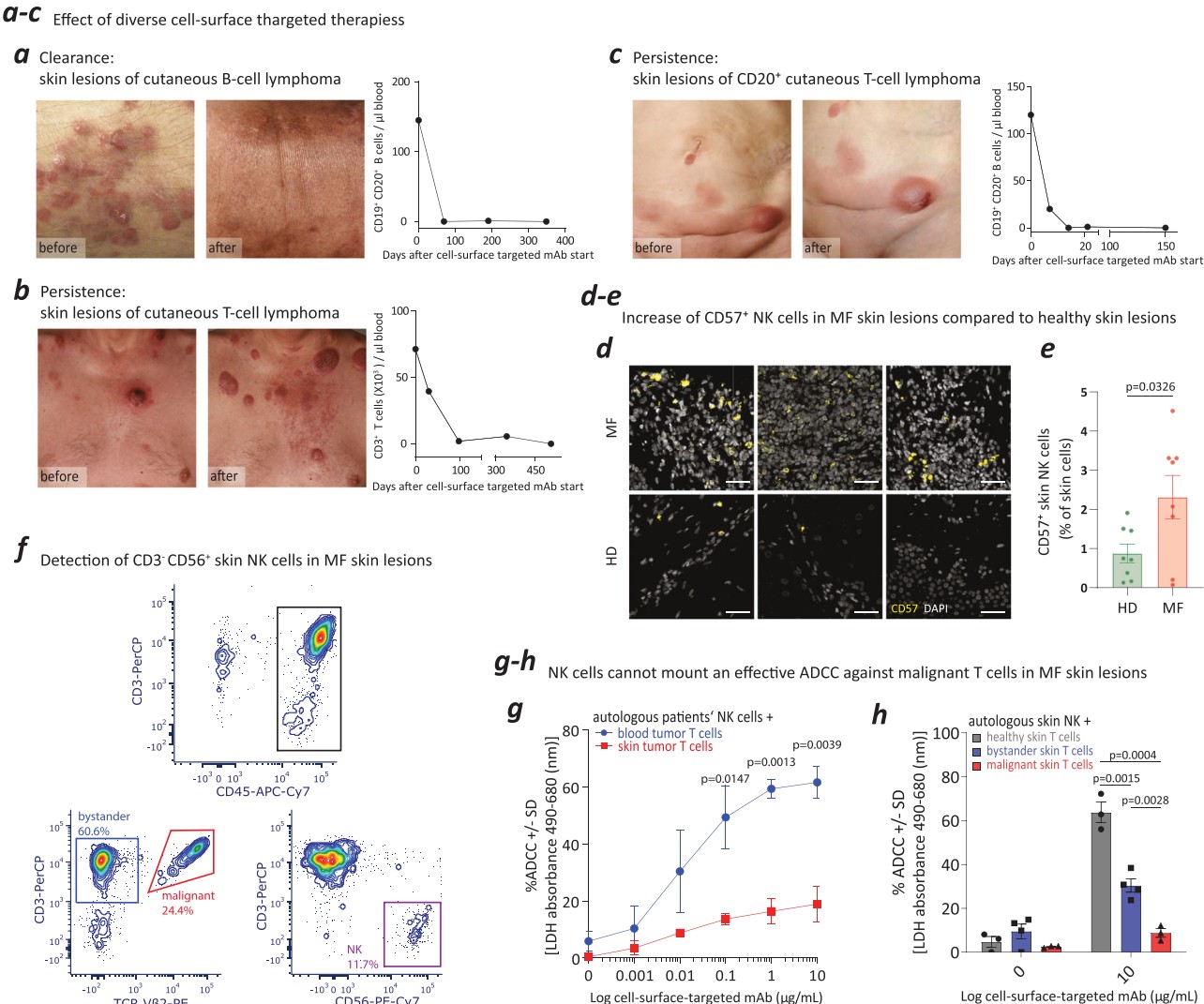

**Fig. 3 | MHC-I negatively impacts NK cell-mediated ADCC activity against malignant T cells in the skin. a** Cutaneous B-cell tumors respond to anti-CD20-mAb (anti-CD20–targeted therapy), which also completely depletes B cells in the blood. **b** Cutaneous T-cell tumors in MF resist anti-CD52-mAb, while CD52+ lymphocytes in the blood are completely depleted. **c** Cutaneous T-cell tumors with aberrant CD20 expression in a patient with MF resist anti-CD20–targeted therapy with anti-CD20-mAb, although CD20+ B cells in the blood are completely depleted. **d** CD57+ NK cells detection in MF and healthy skin lesions. Immunofluorescence staining was performed on MF skin lesions ($N = 7$ MF patients; 3 images were present in the figure) and healthy skin lesions ($N = 7$ healthy individuals; 3 images were present in the figure). Scale bar = 50 μm. **e** Increased percentage CD57+ NK cells in MF skin lesions ($N = 7$) compared with healthy skin ($N = 7$). The calculation was based on immunofluorescence staining in **d**. Data were presented as mean values +/− SEM. The p values were calculated using unpaired, two-tailed student's t test. **f** Skin cells were isolated and gated on the CD45+ lymphocyte population from

T-lymphoma skin lesions of MF patients. Malignant and bystander T cells from T-lymphoma skin lesions were further separated, based on patient T-lymphoma-specific TCR Vβ antibody. The CD3− and CD56+ NK cell populations can be clearly identified in the skin lesions of patients with MF by flow cytometric analysis (representative example of $N = 13$ samples). **g** LDH release detection showed that autologous blood NK cells mount effective ADCC against blood tumor T cells from patients with L-CTCL ($N = 3$) but could not mount an ADCC response sufficient to effectively kill skin tumor T cells from patients with MF ($N = 3$). Data were presented as mean values +/− SEM. The p values were calculated using unpaired, two-tailed student's t test. **h** LDH release detection showed that autologous skin NK cells triggered effective ADCC against healthy skin T cells ($N = 3$) and bystander T cells in MF skin lesions ($N = 4$) but could not mount an effective ADCC against malignant T cells in MF skin lesions ($N = 3$). Data were presented as mean values +/− SEM. The p values were calculated using unpaired, two-tailed student's t test. mAb, monoclonal antibody. Source data are provided as a Source Data file.

(mouse MHC-I). In our study, we focused on Ly49-C/I, which are distinct inhibitory receptors on NK cells in mice.

We performed both in vitro and in vivo experiments using EL4-hCD20, a murine T-lymphoma cell line that was genetically modified to express human CD20, luciferase, and green fluorescent protein. We used human CD20 as the antigen for anti-hCD20 targeted therapy, after verifying its stable expression on EL4-hCD20 cells (Supplementary Fig. 8a). With characteristics similar to MF malignant T cells, EL4-hCD20 showed higher expression of H-2K$^b$ (mouse MHC-I) compared to blood and splenic T cells from C57BL/6 mice (Fig. 5a). Inhibitory Ly49-C/I blockade increased NK cell-mediated anti-hCD20 ADCC

activity against EL4-hCD20, as evidenced through in vitro assessments involving flow cytometry and LDH release detection (Fig. 5b).

Next, we validated these outcomes in murine models in vivo. Subcutaneous inoculation of EL4-hCD20 tumors was performed to replicate the presence of tumor T cells observed in MF skin lesions. A tumor growth assessment established that 100,000 tumor cells represented the optimal quantity for subsequent in vivo investigations (Supplementary Fig. 8b). The therapeutic anti-hCD20-mIgG2a antibody was administered once weekly for 3 weeks, while the blocking anti-Ly49-C/I F(ab')2 antibody was administered twice weekly for the same duration (Fig. 5c). Isotype antibodies, specifically mouse IgG2a

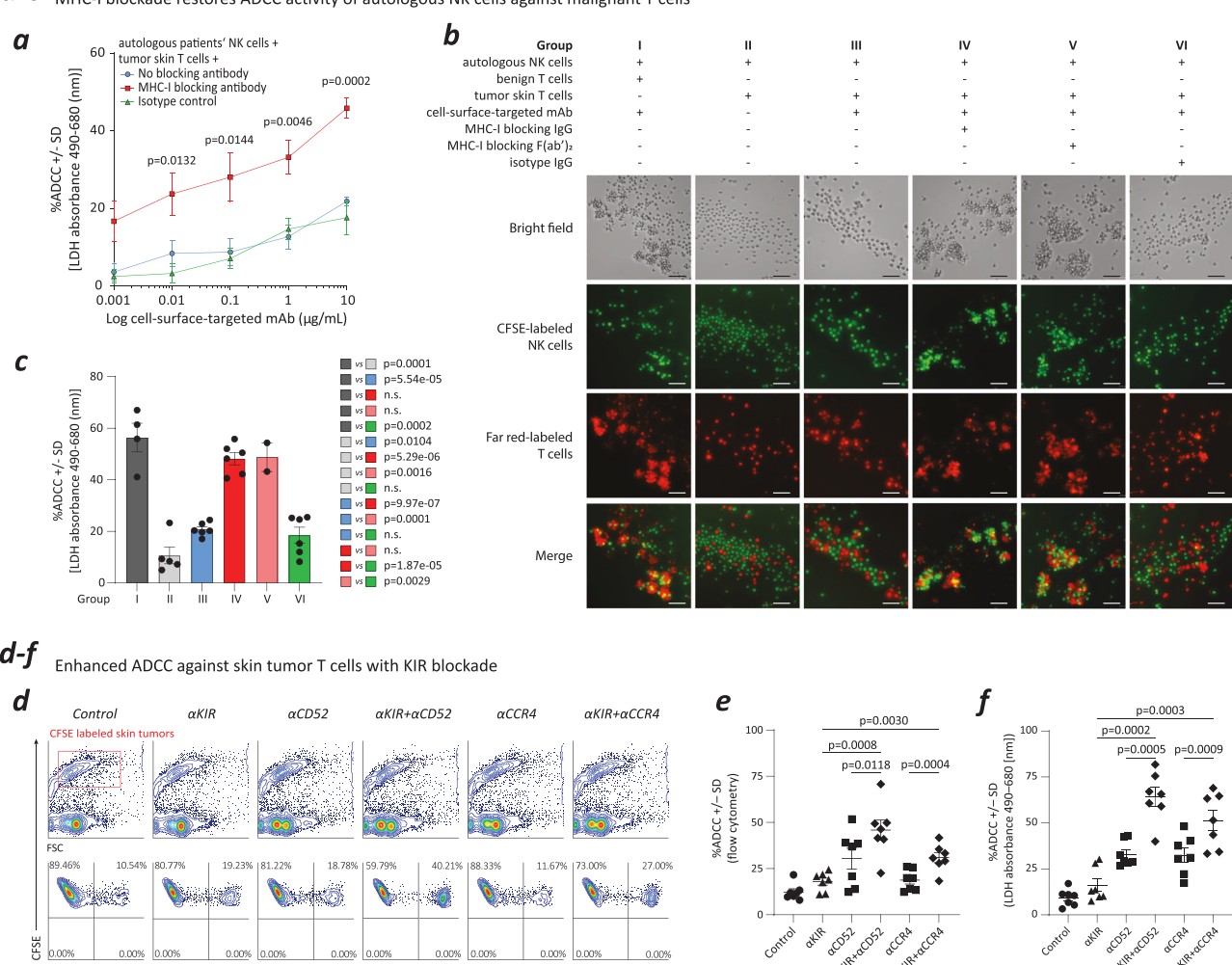

**Fig. 4 | MHC-I blockade restores NK cell-mediated ADCC activity against malignant T cells in MF skin lesions. a** MHC-I blockade restored ADCC activity of autologous NK cells against malignant T cells in MF tumor skin lesions (*N* = 4) when cell-surface receptor-specific targeted mAb (CD52-mAb) was used as an opsonizing agent. Data were presented as mean values +/− SEM. The *p* values were calculated using unpaired, one-way ANOVA test. **b**, MHC-I blockade resulted in ADCC cluster formation, showing that opsonized malignant T cells from tumor skin lesions were surrounded by autologous NK cells. NK cells and T cells were labeled with CFSE and Far Red, respectively. Scale bar = 50 μm. **c**, The corresponding ADCC activity from each group of cells in (**b**), demonstrating restored NK cell-mediated ADCC activity against malignant T cells in MF skin lesions upon MHC-I blockade. *N* = 4 in group I; *N* = 5 in group II; *N* = 6 in group III; *N* = 6 in group IV; *N* = 2 in group V; *N* = 6 in group VI. Data were presented as mean values +/− SEM. The *p* values were calculated using unpaired, two-tailed student's *t* test. n.s., not significant. **d** FACS analysis revealed KIR blockade enhanced ADCC activity of autologous NK cells against malignant T cells from MF skin lesions when anti-CD52-mAb or anti-CCR4-mAb was used as the opsonizing agent. **e**, Flow cytometrical detection showed enhanced ADCC against skin tumor T cells when KIR was blocked (*N* = 7). Data were presented as mean values +/− SEM. The *p* values were calculated using paired, two-tailed student's *t* test. **f** LDH release detection showed increased ADCC against skin tumor T cells when KIR was blocked (*N* = 7). Data were presented as mean values +/− SEM. The *p* values were calculated using paired, two-tailed student's *t* test. Source data are provided as a Source Data file.

and IgG2a F(ab′)2, were used as controls. We employed an in vivo imaging system (IVIS) to track tumor progression through luminescence intensity measurement of EL4-hCD20 tumors. Prior to treatment, baseline assessment on day 3 validated successful inoculation of mice with EL4-hCD20 tumors, and luminescence signals exhibited a comparable level across all groups (Fig. 5d; Supplementary Fig. 8c). By day 9 post-treatment, EL4-hCD20 tumors were exclusively localized to the skin. Notably, tumors in the two control groups exhibited well-established growth, characterized by consistent and intense luminescence signals.

Monotherapy with either the anti-Ly49-C/I blocking antibody or the therapeutic anti-hCD20 antibody individually resulted in a trend toward slower tumor growth. As anticipated based on the in vitro findings, the combination of the ADCC-inducing therapeutic antibody and the NK-cell immune checkpoint Ly49-C/I

blocking antibody led to enhanced tumor suppression, paralleled by a significant reduction in luminescence intensity (Fig. 5d; Supplementary Fig. 8c). Concurrently, the combination treatment exhibited a delayed tumor growth (Fig. 5e), accompanied by a reduction in tumor volume (Supplementary Fig. 8d, e) and an enhancement in overall survival when contrasted to monotherapy and control treatments (Fig. 5f).

**Autocrine IL-32β acts as a stimulator for MHC-I expression in malignant T cells**

Cytokines are one of the key factors contributing to MHC-I induction. Our NN-log-reg method identified 15 cytokine-related genes that were consistently deemed significant for identifying malignant T cells across both the training and test datasets (Supplementary Fig. 2g). By analyzing the expression of those 15 cytokine-related genes, we found that

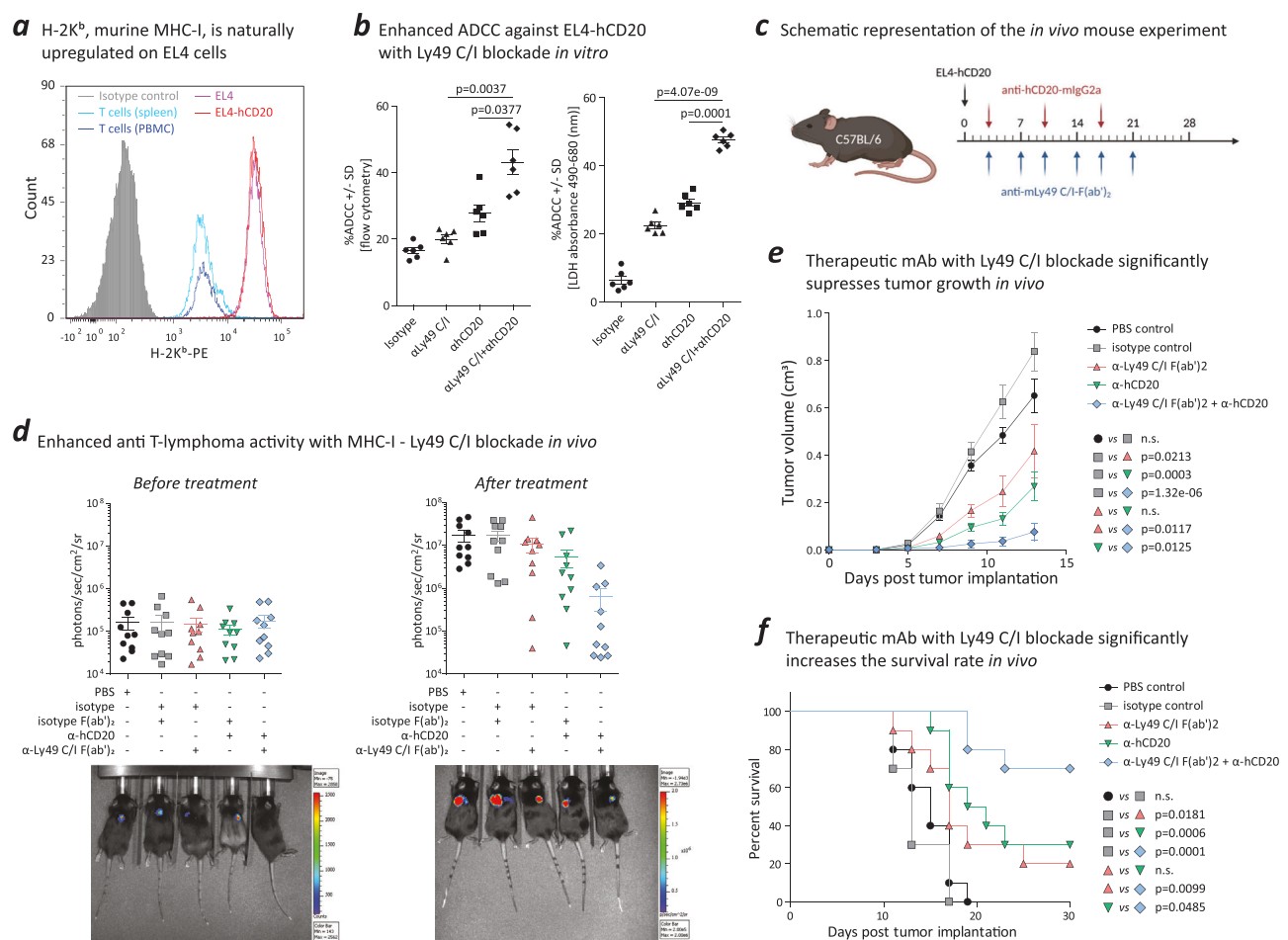

**Fig. 5 | Blockade of the MHC-I−KIR interaction enhances NK cell-mediated anti-T-cell lymphoma activity in mice in vitro and in vivo. a** The murine T-lymphoma cell line, EL4-hCD20, highly expresses H-2K$^b$ versus the normal T cells of wild-type C57BL/6 mice. H-2K$^b$ is the MHC class I molecule for C57BL/6 mice. **b** Two independent sets of experiments, one based on flow cytometry and one based on LDH release detection, indicated increased ADCC against murine T-lymphoma cells during Ly49 C/I blockade. Ly49 C and Ly49 I are specific inhibitory receptors on NK cells in mice ($N = 6$). Data were presented as mean values +/− SEM. The $p$ values were calculated using paired, two-tailed student's t test. **c** Schematic representation of the in vivo mouse experiment. C57BL/6 mice were inoculated subcutaneously with $1 \times 10^5$ EL4-hCD20 T-lymphoma cells. After tumor inoculation, mice received 250 μg of therapeutic anti-hCD20-mIgG2a or isotype IgG2a control on day 3 and once weekly for three weeks, 200 μg of blocking anti-Ly49 C/I F(ab')$_2$ or isotype IgG2a F(ab')$_2$ on day 3 and biweekly for three weeks, or a combination of anti-hCD20-mIgG2a and anti-Ly49 C/I F(ab')$_2$. Therapeutic and blocking antibodies were administered i.p. The schematic representation is created with BioRender.com. **d** Tumor tracking and intensity of luminescence signals from tumor cells as measured by IVIS ($N = 10$). On day 3 before treatment, the mice were confirmed to have successful EL4-hCD20 tumor inoculation, and the intensity of luminescence signals was approximately equal in each group. On day 9 after treatment, tumor cells remained only in the skin and no metastasis had occurred. In addition, the intensity of luminescence signals indicated that Ly49 blockade enhances the anti-T−cell lymphoma activity of anti-hCD20 therapeutic antibody. Data were presented as mean values +/− SEM. **e** The tumor growth curve illustrated that both anti-hCD20 therapeutic antibody and Ly49 C/I F(ab')$_2$ blocking antibody were able to inhibit tumor growth in the skin, and the combination of both antibodies significantly inhibited tumor growth in the skin ($N = 10$). Data were presented as mean values +/− SEM. The $p$ values were calculated using unpaired, two-tailed student's $t$ test from the data on day 13. n.s., not significant. **f** The combination of anti-hCD20 therapeutic antibody and Ly49 C/I F(ab')$_2$ blocking antibody against T-cell lymphoma in the skin significantly increased overall survival ($N = 10$). The $p$ values were calculated using simple survival analysis (Kaplan-Meier). n.s., not significant. Source data are provided as a Source Data file.

malignant skin T cells can be distinguished from non-malignant T cells by their significantly increased expression of *IL32* and decreased expression of *IL7R* (Fig. 6a; Supplementary Fig. 9). Using the transcript per million results from scRNA-seq, we identified *IL32β* as the predominant isoform expressed in malignant T-cell populations in the skin (Fig. 6b).

A noteworthy observation was the positive correlation between the expression of classical *MHC-I* and *IL32* in tumor skin T cells, a connection not observed in single bystander or healthy T-cell populations (Fig. 6c). Furthermore, NanoString analysis revealed a parallel increase in levels of classical *MHC-I* and *IL32*, which positively correlated with the proportion of the malignant TCR clone in MF tumor skin lesions. This relationship was not evident in skin lesions from atopic dermatitis, a benign inflammatory skin condition (Fig. 6d).

Additionally, we conducted Western blot analysis to assess the protein expression level of IL-32 in T cells from both healthy skin and MF skin lesions. IL-32 protein exhibited significant upregulation on MF skin T cells in comparison to healthy skin T cells (Fig. 6e).

To demonstrate the relevance of our findings, we suppressed IL-32 expression by transfecting *IL32* small inhibitory RNA (siIL32) into My-La CD4$^+$ cells, an MF-CTCL cell line. The siIL32 transfection not only decreased the expression of *IL32*, the expression the classical MHC-I molecules also decreased (Fig. 6f). In addition, siIL32 transfection decreased the cell viability of My-La CD4$^+$ cells (Fig. 6g), in line with its recently reported role as an important CTCL tumor survival factor[45].

While IL-32 has been shown to elicit various cellular responses and influence cancer and inflammatory processes, the specific receptor that directly binds and interacts with IL-32 has not yet been definitively

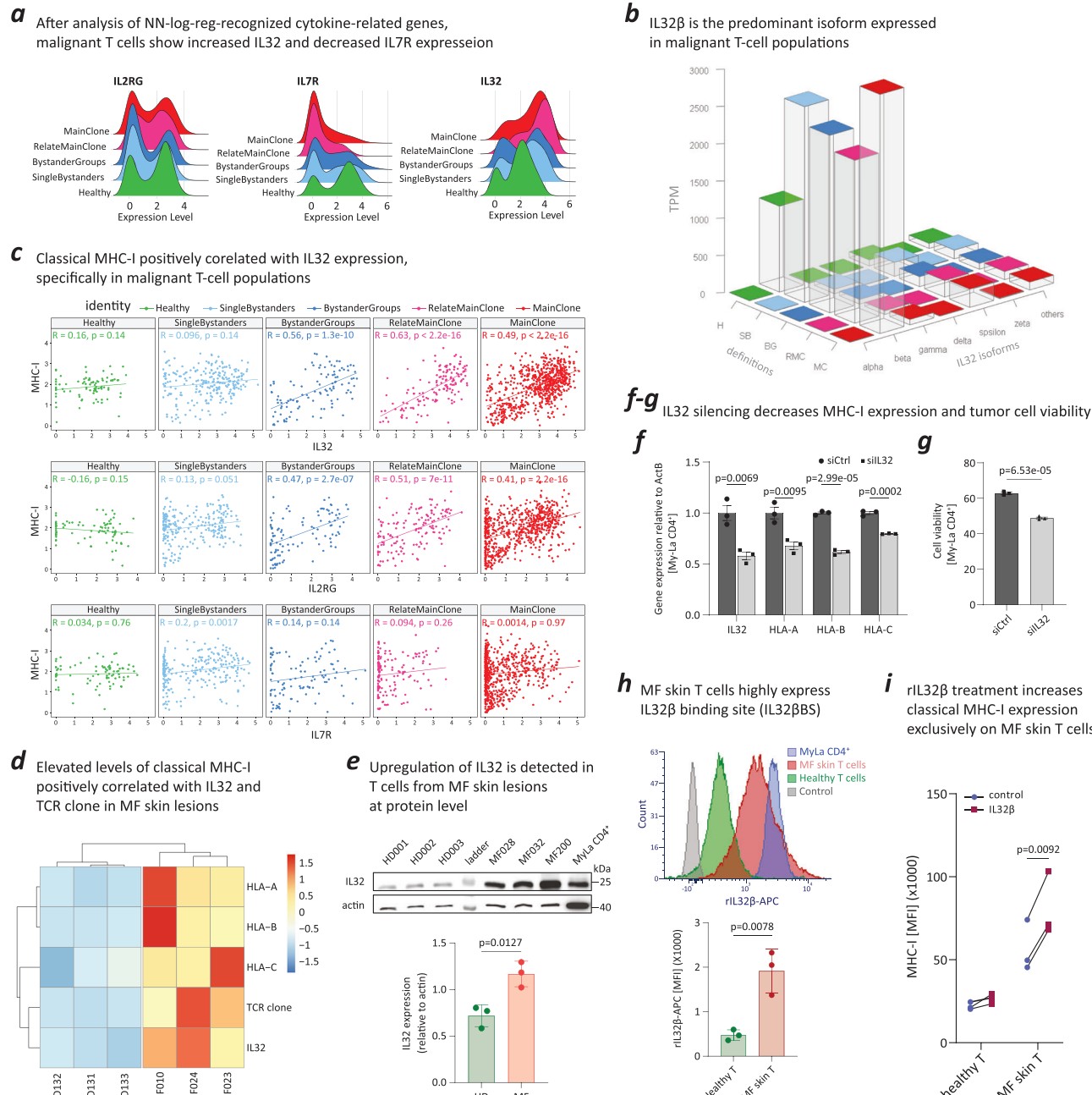

**Fig. 6 | Malignant T cells secret IL-32β isoform to promote classical MHC-I expression on malignant T cells. a–c** Analyzing cytokine-related genes ($N = 526$ cells for Main Clone (MC); $N = 141$ cells for Relate Main Clone (RMC); $N = 50$ cells for Bystander Groups (BG); $N = 108$ cells for Single Bystanders (SB); $N = 85$ cells for Healthy (H)). **a** Malignant T-cell populations show significantly increased *IL32* expression and decreased IL7R expression. **b** *IL32β* is the predominant *IL32* isoform expressed in malignant T-cell populations. **c** *MHC-I* expression positively correlated with *IL32* expression, specifically in malignant T-cell populations but not in single bystander T-cell or healthy T-cell populations. The *p* values were calculated using correlation test. **d** NanoString analysis showed that elevated levels of *MHC-I* molecules and *IL32* are positively correlated with the proportion of TCR clonal T cells in the micro-environment of MF skin lesions ($N = 3$) but not in benign inflammatory skin disease ($N = 3$). The color scale indicates higher gene expression in red and lower gene expression in blue. **e** IL-32 upregulated at the protein level in T cells from MF skin lesions ($N = 3$). Data were presented as mean values +/− SEM.

The *p* values were calculated using unpaired, two-tailed student's *t* test. siIL32 RNA transfection into My-La CD4$^+$ cells. **f** RNA interference knockdown of *IL32* resulted in decreased expression of IL-32 and three classical MHC-I molecules (HLA-A, HLA-B and HLA-C) in My-La CD4$^+$ cells ($N = 3$), analyzed by qRT-PCR. Data were presented as mean values +/− SEM. The *p* values were calculated using unpaired, two-tailed student's *t* test. **g** siIL32 decreased the viability of My-La CD4$^+$ cells ($N = 3$). Control siRNA served as a negative control. Data were presented as mean values +/− SEM. The *p* values were calculated using unpaired, two-tailed student's *t* test. **h** T cells from MF skin lesions show high expression of IL-32β binding sites (IL-32βBS) on their cell surface ($N = 3$). Data were presented as mean values +/− SEM. The *p* values were calculated using unpaired, two-tailed student's *t* test. **i**, Increased classical MHC-I expression exclusively on T cells from MF skin lesions in response to IL-32β stimulation ($N = 3$). The *p* values were calculated using paired, two-tailed student's *t* test. Source data are provided as a Source Data file.

identified. To assess the presence of the IL-32β binding sites (IL-32βBS) on MF skin T cells, we labeled recombinant IL-32β (rIL-32β) with allophycocyanin (APC) and incubated the APC-labeled rIL-32β with healthy T cells, MF skin T cells, and the My-La CD4⁺ MF-CTCL cell line. Both MF skin T cells and My-La CD4⁺ MF-CTCL cells exhibited higher levels of conjugated APC-labeled rIL-32β, underscoring a significant upregulation of IL-32β binding sites on MF skin T cells in comparison to healthy T cells (Fig. 6h). Furthermore, the addition of rIL-32β resulted in a selective augmentation of classical MHC-I expression specifically in MF skin T cells, while leaving healthy T cells unaffected (Fig. 6i). This underscores the significance of the IL-32β - MHC-I axis in MF skin T cells, implying that IL-32β functions as a crucial autocrine signal for survival and expansion of malignant T cells, and a critical determinant in tumor T-cell immune evasion within the human skin environment.

## Discussion

Cancer is associated with progressive impairment of cellular immunity. In MF, the most common type of cutaneous T-cell lymphoma, despite the favorable response of blood disease to treatment, tumoral skin lesions in the majority of patients exhibit resistance to therapeutic mAbs targeting tumor cell-surface antigens. In the present study, the NN-logreg model reveal that MHC-I, also known as HLA-I in humans, is overexpressed on tumor skin T cells, leading to the inhibition of NK cell activity and NK-mediated ADCC, thereby conferring resistance to targeted therapies. Blocking MHC-I and inhibitory KIR (inhibitory Ly49 murine homologue) restores ADCC and enhances the antitumoral activity of therapeutic mAbs, both ex vivo in humans and in vitro/in vivo in a murine T-cell lymphoma model, resulting in reduced tumor volumes and significantly increased survival rates. Consequently, we propose a tumor immune escape mechanism that accounts for impaired ADCC due to the increased expression of MHC-I on tumor skin T cells.

The use of artificial intelligence (AI) and ML algorithms in medicine has great potential to facilitate accurate diagnosis, disease stage prediction, and discovery of biomarkers[46–48]. In CTCL, an AI and ML algorithm modeling the distinct transcriptomic states within L-CTCL has been developed with almost 80% accuracy in disease-stage prediction[6]. A recent study addressed the combination of label-free imaging and a weakly supervised deep learning approach for blood diagnostics in L-CTCL[49]. The ever-increasing use of scRNA-seq has inevitably led to an increasing application of ML and deep learning methods to analyze these data. Recently, these methods were used for batch-effect removal and unsupervised clustering of cells[50], data imputation[51] and disease stage prediction[52]. The model for the mathematical reconstruction of T-cell receptors (TraCeR) was developed from single-cell transcriptomes[53] and is a powerful tool to use in studies of T-cell malignancies; nevertheless, there may still be a small chance to define bystander T cell as malignant if bystander T cell shares exactly the same TCR as malignant T cells.

In the specific context of our research, deep learning methods were best suited for analyzes of our gene expression dataset[54]. This dataset comprises a limited number of samples but includes numerous features. We conducted deep sequencing exclusively on skin T cells, resulting in small sample numbers but high coverage of sequenced genes. This approach allowed ML methods to excel in identifying features that distinguish the two T-cell subpopulations, namely tumor and non-tumor T cells. Furthermore, the NN approach successfully captured all relevant genes, whether they were upregulated or downregulated. This unbiased analysis facilitated the exploration of complex transcriptome patterns through ML techniques, which are ideal for handling high-dimensional data with nonlinear relationships. Subsequently, we performed traditional statistical and functional analyses, further enhancing our understanding of tumor T cells in CTCL. This approach also enabled us to directly analyze the transcriptome of both the malignant T-cell clone and bystander T cells, providing valuable intra-individual control information.

In general, upregulated MHC-I expression is related to a favorable prognosis in solid tumors; however, tumor upregulation of classical MHC-I has been shown in vivo to inhibit NK cells[55]. Further, MF skin tumor T cells have significantly increased MHC-I expression compared to healthy CD4⁺ T cells[56]. We discovered that the overexpression of classical MHC-I on tumor T cells plays a pivotal role in immune evasion within cancer, explaining the observed reduction in NK cell-mediated ADCC in MF skin. These findings hold significant relevance in the context of contemporary targeted therapies and immunotherapies utilizing NK cells for anticancer treatment. A recent publication by Scheffschick et al. further underscores the importance of NK cells in skin T cell lymphoma lesions, offering additional evidence regarding their presence and altered phenotype[57].

Previous studies have reported that IL-32 is highly expressed in the skin lesions of patients with MF and contributes to the survival of malignant T cells in CTCL[45,58]. Moreover, the IL-32β isoform is produced by circulating T cells in L-CTCL[59]. In our current study, we observed that malignant skin T cells display elevated levels of IL-32β binding sites, and the supplementation of IL-32β leads to an increase in MHC-I expression. The IL-32β produced by malignant T cells thus functions as an autocrine signal, enhancing MHC-I expression and thereby prolonging the survival of malignant T cells in MF skin lesions. Consequently, we believe that the neutralization of IL-32β may prove to be another crucial factor in supporting the efficacy of NK cell-mediated antitumor therapies in MF.

Given the dependence of ADCC on the ratio of NK cell-activating versus cell-inhibitory signals, a dual combination of tumor cell-surface antigen-targeted mAbs and NK inhibition-blocking mAbs acting as NK-immune checkpoint inhibitors may be a highly promising option for individualized cancer treatment[23]. Recently, blocking the NKG2A immune checkpoint by engaging HLA-E (nonclassical MHC-I) has proven effective in enhancing the anti-metastatic functions of NK cells[24,60]. Specifically, in cases such as pancreatic ductal adenocarcinoma, Monalizumab (IPH2201), a human IgG4 blocking monoclonal antibody targeting inhibitory NKG2A on NK cells and a subset of CD8⁺ T cells, disrupts the interaction between inhibitory NKG2A and HLA-E. This disruption not only strengthens NK-mediated ADCC-inducing anti-tumor responses but also enhances the effector functions of CD8⁺ T cells and prevents liver metastasis[25].

In CTCL, our findings show that all three classical MHC-I proteins (HLA-A, HLA-B, and HLA-C), but not the non-classical MHC-I (HLA-E), exhibit high expression levels on tumor T cells from MF skin lesions. Consequently, we posit that lirilumab may offer more significant potential advantages in augmenting the effectiveness of NK cell-mediated tumor-cell-surface-antigen targeted therapy in MF. Lirilumab (IPH2102/BMS-986015), a fully human IgG4 monoclonal antibody that targets inhibitory KIR2DL1/2/3 expressed on NK cells, is employed to disrupt the interaction between inhibitory KIRs and HLA-C (a classical MHC-I). In clinical settings, lirilumab has demonstrated the ability to elevate NK-mediated antitumor activity and enhance tumor cell-surface antigen-targeted therapy[22,23,61]. This observation suggests that lirilumab may hold promise in enhancing NK cell-mediated tumor-cell-surface-antigen targeted therapy in MF as well.

Further, the first-in-class anti-KIR3DL2/CD158k humanized NK-mediated ADCC-inducing antibody lacutamab (IPH4102), designed to selectively destroy CTCL cancer cells, presents a compelling option[62]. KIR3DL2 is preferentially expressed on malignant T cells and accounts for enhanced resistance to activation-induced cell death in CTCL[63]. Based on this, the main mode of action of lacutamab in CTCL is currently believed to be a NK cell-mediated ADCC effect on KIR3DL2-expressing tumor T cells[64–66]. Additionally, KIR3DL2 belongs to the KIR group of NK cell inhibitory receptors, and as such is an ideal target for NK cell-based checkpoint blockade.

The preferential targeting of malignant T cells by lacutamab and its potential to restore NK cell functions make it an option in the treatment

of CTCL. It is possible that targeting KIR3DL2 with anti-KIR3DL2 mAbs (aKIR3DL2 mAbs) may not only directly affect tumor T cells but circumvent an important cancer immune-suppressive mechanism by restoring NK cell functions. Early results from a phase II trial have shown promising response rates in the skin of patients with KIR3DL2-expressing advanced MF, reaching 57.1%. These response rates surpass those typically observed with other therapeutic monoclonal antibodies targeting T-cell derived markers, which further underscores the translational potential of our findings in clinical practice[67].

In summary, while these findings are promising, more research is needed to fully realize the potential of these approaches. Nonetheless, they provide insights into the ongoing efforts to advance cancer treatment and may pave the way for personalized therapies in the future.

## Methods

All experiments in this study were conducted in accordance with the principles of the Declaration of Helsinki, and the study design was approved by the Institutional Review Board of the University of Zurich (KEK-ZH-Nr. 2015-0209).

### Collection of human skin and blood samples

All patients in this study met diagnostic criteria for early-stage MF according to the tumor/node/metastasis system and stage classification (Supplementary Table 1). The informed consent signed by each individual patient for collecting samples and publishing clinical information potentially identifying individuals was obtained. The authors affirm that human research participants provided informed consent for publication of the images in Fig. 3a–c. Patient skin biopsies, serum and peripheral blood samples, as well as discarded tissue from surgical remnants, were obtained from the University of Zurich Biobank (EK No. 647) and from the VITA certified Dermatology Biobank (CHUV_2103_12) of the Lausanne University Hospital (CHUV). Blood and serum samples from healthy individuals were obtained anonymously from the blood banks of the University Hospital of Zurich and the Lausanne University Hospital (CHUV), and informed consent was obtained from donors.

### Cells, antibodies and reagents

Peripheral blood mononuclear cells (PBMCs) were separated from the whole blood of patients and healthy individuals using Ficoll-Paque density gradient centrifugation (GE Healthcare, 17-1440-03), and NK cells were isolated using CD56 MicroBeads (Miltenyi Biotec, 130-050-401) according to the manufacturer's instructions. Skin T cells were isolated from skin biopsies of patients with MF and healthy individuals using collagen-coated CellFoam matrices and cultured in Iscove's Modified Dulbecco's Medium (Thermo Fisher, 12440-053), supplemented with 20% Gold FBS (PAA Laboratories, A15-151), 1× antibiotic-antimycotic (Thermo Fisher, 15240-062), 2 mM L-glutamine (Biochrom, K0282), 100 IU/ml IL2 and 10 ng/ml IL15 (Peprotech, 200-15)[12]. BCC, SCC and CBCL cells were isolated from skin biopsies using Liberase (final concentration 0.5 mg/ml, Roche 54001020001) and incubated at 37˚C for 1 h.

Raji (ATCC-CCL-86), a Burkitt's lymphoma-derived B-cell line, was maintained in complete RPMI 1640 culture medium containing 2 mM L-glutamine, 1 mM sodium pyruvate, 1× antibiotic and 10% Gold FBS. Ramos (RA1, ATCC-CRL-1596), a Burkitt's lymphoma-derived B-cell line, was purchased from LGC Standards and maintained in modified RPMI 1640 culture medium containing 2 mM L-glutamine, 10 mM HEPES, 1 mM sodium pyruvate, 4500 mg/L glucose, 1× antibiotic and 10% Gold FBS. My-La CD4+ (ECACC, catalog no. 95051032), an MF cutaneous T-cell lymphoma cell line, was purchased from Merck and cultured in RPMI 1640 culture medium containing 2 mM L-glutamine, 10 U/ml IL-2, 10 U/ml IL-4, 1× antibiotic and 10% human AB serum. EL4 (ATCC TIB-39), a mouse T-cell lymphoma-induced in a C57BL/6 mouse, was cultured in complete RPMI 1640 medium. EL4-hCD20 (Clone 1E12A5), EL4 mouse T-lymphoma cells transduced with hCD20, luciferase and green fluorescent protein-cDNA, was provided by Dr

Jeanette Leusen (Laboratory for Translational Immunology, UMC Utrecht, the Netherlands) and cultured in complete RPMI 1640 medium.

Anti-CD20-mAb (a chimeric human IgG1; MabThera® 10 mg/ml, Roche Pharma [Schweiz] AG) and anti-CD52-mAb (human IgG1; Lemtrada® 30 mg/ml, Sanofi-Aventis) were obtained from the Cantonal Pharmacy of Zürich, Switzerland. Anti-CCR4-mAb (human IgG1; Gly-007CL) was purchased from Creative Biolabs. Anti-KIR-mAb, a fully human IgG4 that targets KIR2D (lirilumab; HY-P99208-1MG), was ordered from MedChemExpress. Commercially available monoclonal blocking antibody against human MHC-I (clone B9.12.1; Beckman Coulter, Krefeld, Germany), along with its F(ab')2 fragment and isotype control (mouse IgG2a; BioLegend), were used at a concentration of 10 μg/ml for the MHC-I blockade experiments. Anti-MHC-I F(ab')2 fragments were prepared with the Pierce F(ab')2 Micro Preparation Kit (Thermo Fisher, 44688) and verified by nonreducing sodium dodecyl sulfate polyacrylamide gel electrophoresis (SDS-PAGE). Monoclonal mouse IgG2a antibody against human CD20, anti-hCD20-mIgG2a (hcd20-mab10, InvivoGen), which features the constant region of the mouse IgG2a isotype and the variable region of anti-CD20-mAb, was used as a therapeutic antibody in in vitro and in vivo mouse experiments. For in vitro experiments, anti-hCD20-mIgG2a was used at a concentration of 10 μg/ml; for in vivo experiments, 250 μg of anti-hCD20-mIgG2a was administered intraperitoneally (i.p.) once weekly for three weeks. An equal amount of mouse IgG2a isotype control (BioLegend, 401504) was used in control groups. The F(ab')2 fragments of mouse monoclonal antibody against mouse Ly49 C and Ly49 I (mouse IgG2a, clone 5E6; BD Biosciences) and isotype mouse IgG2a (clone G155-178, BD Biosciences) were also used in vivo. Anti-Ly49 C/I F(ab')2 was used in vitro as a blocking antibody at a concentration of 10 μg/ml, and 200 μg of blocking anti-Ly49 C/I F(ab')2 was administered i.p. twice weekly for three weeks in in vivo experiments. An equal amount of mouse IgG2a isotype F(ab')2 was used in control groups.

### Flow cytometry

For flow cytometry analysis, cells were collected, washed and resuspended in 50 μl of ice-cold magnetic-activated cell sorting (MACS) buffer (phosphate-buffered saline [PBS, pH 7.2], 0.5% bovine serum albumin, and 2 mM ethylenediaminetetraacetic acid) and fluorescent-conjugated antibodies for 20 min on ice, followed by two washes with MACS buffer.

Analysis of PBMCs and skin cells of patients and healthy donors was performed using monoclonal antibodies against human CD3 (clone BW264/56; label PerCP; Miltenyi Biotec, 130-113-131), CD3 (clone: SK7; label FITC; Invitrogen, 11-0036-42), CD3 (clone: HIT3a; label PerCP/Cyanine5.5; Biolegend, 300328), CD4 (clone VIT4; label APC-Vio770; Miltenyi Biotec, 130-098-153), CD4 (clone: RPA-T4; label APC/Cyanine7; Biolegend, 300518), CD8 (clone BW135/80; label PE-Cy7; Miltenyi Biotec, 130-096-556), CD8 (clone: SK1; label PE/Cyanine7; Biolegend, 344712), CD16 (clone VEP13; label APC; Miltenyi Biotec, 130-091-246), CD20 (clone LT20; label FTIC; Miltenyi Biotec, 130-098-081), CD45 (clone 5B1; label APC-Vio770; Miltenyi Biotec, 130-096-609), CD45 (clone HI30; label PerCP-Cyanine5.5; eBioscience, 45-0459-41), CD56 (clone AF12-7H3; label PE-Vio770; Miltenyi Biotec, 130-096-831), TCR Vβ (clone ZOE; label PE; Beckman Coulter), HLA-A,B,C (clone REA230; label FITC; Miltenyi Biotec, 130-101-447), HLA-A,B,C (clone: W6/32; label APC; Biolegend, 311410) and HLA-E (clone: 3D12; label Brilliant Violet 421; Biolegend, 342612). Isotype-matched negative control antibodies were used to set the gates for positive staining. Vβ clonal T-cell populations were assessed by flow cytometry using the IOTest® Beta Mark TCR Vβ Repertoire Kit (Beckman Coulter, IM3497).

Analysis of PBMCs and splenocytes from wild-type C57BL/6 mice and the EL4, EL4-hCD20 and Raji cell lines was performed using monoclonal antibodies against mouse CD3 (clone 17A2; label PerCP/Cyanine5.5; BioLegend, 100217), CD4 (clone RM4-5; label APC-Vio770;

BioLegend, 100525), CD8 (clone 53-6.7; label PE-Vio770; BioLegend, 100721), CD45 (clone 30-F11; label APC; BioLegend, 103111), H-2K$^b$ (clone AF6-88.5; label PE; BioLegend, 116507) and APC-conjugated anti-hCD20-mIgG2a (APC conjugation kit; Abcam, ab201807).

Validation statements and dilutions of all antibodies used for flow cytometry were described on the manufacturer's website with relevant citations. The stained cells were acquired at least 10,000 cells per sample on Becton Dickinson FACSCanto™ and LSRFortessa™ instruments, and data were analyzed using FCS Express 7 Flow Cytometry RUO (De Novo Software) and Prism v9.1.0 (GraphPad) software.

## ScRNA-seq, quality control, gene quantification and data analysis

All samples for scRNA-seq were collected between 2013 and 2017 (Supplementary Table 1) and were processed between 2016 and 2018. Capture and processing of single skin T cells was performed using the Fluidigm C1 Autoprep system. Cells were loaded at a concentration of 2,000 cells/μl onto C1 integrated fluidic circuit chips for 5–10 μm cells. All C1 capture sites were microscopically inspected to identify the sites that contained only a single cell. Empty sites and those with multiple cells were excluded from further analysis. External RNA Controls Consortium spike-in RNAs served as a control. The SMARTer® Ultra® Low RNA Kit (Clontech) was used for reverse transcription and cDNA pre-amplification. The single-cell cDNA products from each cell were then used to prepare Illumina sequencing libraries and sequenced as paired-end 150-base reads on the Illumina HiSeq 4000 platform, which was provided by the Functional Genomic Center Zürich.

Quality control was performed using FastQC v. 0.11.7 (Babraham Bioinformatics, http://www.bioinformatics.babraham.ac.uk/projects/fastqc/), and the adaptors and low-quality bases with a Phred quality score <20 were trimmed from the ends of the reads using Trim Galore v. 0.4.4 (Babraham Bioinformatics)[68].

## TCR reconstruction for clonality analysis

Bioinformatics approaches based on scRNA-seq enable the reconstruction of complete TCR recombinants. The TraCeR system was used in this study to reconstruct TCR recombinants independently for every cell, and gene expressions for cells with an identified TCR were quantified with Kallisto v. 0.45.0 (Pachter Lab) as part of the TraCeR workflow. Preliminary, TCR-clonality was evaluated by a modified TraCeR system (https://github.com/pesho-ivanov/celldive/tree/main/src/tracer). The result of this analysis for each skin T cell was a set of sequences in the form '<V locus>_<junction sequence>_<J locus>', where the gene loci are represented by their names and the junction sequence is a short DNA sequence found between the V and J loci. The gene-expression data were normalized with Scater v. 1.10.1 and Scran v. 1.10.2 (both available through Bioconductor [https://www.bioconductor.org/]) using alternative normalization strategies based on counts per million.

## Clonality criteria

We excluded non-TCR-reconstructed cells lacking both reconstructed α and β chains from each sample. TCR-reconstructed cells were required to have at least one TCR chain reconstructed. The clonality criteria for the "main-clone" were defined on a per-patient basis; the largest group of cells sharing the same combination of α and β chains was designated as the "main-clone". For instance, if the most prevalent combination was "α1 with β1", then "main-clone" cells would encompass either α1 or β1 chains in their TCR combination, such as "α1 with β1 (or β2)" or "α1 (or α2) with β1". The "related-to-main-clone" group included cells that did not match the α or β chains of the "main-clone" but had a relative association with "main-clone" cells, like "α2 with β3 (or β4)" or "α3 (or α4) with β2". "Bystander groups" comprised cells with reconstructed chains that neither matched nor had any relative associations with the α or β chains of "main-clone" cells; these cells needed to exhibit at least two cells sharing the same α and β chains within the sample. The

"single bystanders" category encompassed cells lacking reconstructed α or β chains shared with any other cells in the sample.

## ML and neural network architecture

Based on clonality rules, "main-clone" and "related-to-main-clone" cells are considered as clonal cells, and "bystander groups" and "single bystanders" cells are considered bystander cells. T cells without the CD8 phenotype were pre-selected for ML analysis. Only the TCR-reconstructed T cells were exclusively selected for ML analysis, ensuring the robustness and reliability of the analysis.

For ML analysis, all genes related to the variable parts of the TCR (TRAVxx, TRAJxx, TRBVxx, TRBJxx) were excluded. Cells were divided into training, validation, and test sets. A manual hyperparameter search was performed. Models were trained using all available skin T cells. Reads per kilobase million-normalized transcript counts were used for ML analysis; normalized data were used as input, and malignancy was used as target (clonal cells = 1, bystander cells = 0).

We compared logistic regression with L2 regularization[31], support vector machines[31] a gradient-boosted tree-based model (XGBoost[33]), TabNet[32] and a 'standard artificial neural network' (five linear layers, 1000, 1000, 1000, 32 and 1 neurons, respectively, SELU activation functions, last: sigmoid; implemented using pytorch[30]) and an adaptive logistic regression model, for which the model weights were set with a neural network (NN-log-reg, implemented in pytorch[30]). For the NN-log-reg method, a hypernetwork was used to calculate the adaptive weights for an "adaptive logistic regression" analysis.

The input vector for this hypernetwork was the length of the detected genes without TCR genes ($n = 44,782$ genes). The network consists of three linear layers (1000 units each), with two tanh activation functions and a sigmoid (σ) function followed by a softmax function at the end (Fig. 1d). The last layer is the same shape as the input and represents the adaptive weights. After the softmax, each value of this layer is multiplied with the input (logistic regression), whereas the first value of the importance weight matrix represents the weight for the first gene in the gene list, and so on. The sum of these products is followed by a (1,1) linear layer and a sigmoid-activation function. Binary cross-entropy was used to calculate the loss. Mathematically, the method can be described as follows:

$$output = sigmoid\left(bias_D + w_D * \left(hypernet(\boldsymbol{x})^T * \boldsymbol{x}\right)\right)$$

$$where\ hypernet(x) = SoftMax\big(sigmoid\big(bias_C + W_C * \tanh(bias_B \\ + W_B * \tanh(bias_A + W_A(x)))\big)\big)$$

The adaptive weights were used for determining the most important genes for predicting the malignancy of each cell. The feature weights of all cells of both the training and test sets were compared with each other. A transcript was considered as 'important' for malignancy prediction for a cell if it was among the top 0.5% of transcripts according to attention values. Only correctly classified cells were considered. The percentage of cells for which this criterion is true is indicated in the figures (Fig. 1g; Supplementary Fig. 3).

## Clonality diagrams and single-cell gene-expression analysis

The TCRβ chain variable distribution plot was generated using mosaicplot. Seurat (v.3.1)[69] was used to scale and normalize gene-expression data for clustering and differential gene-expression analysis. Gene set enrichment analysis was conducted using gprofiler2[70]. The gene expression of the top pivotal clonal genes on real-time disease stage was performed by using SuperPlots[71].

## Western blotting

Skin T cells (1,000,000 cells) were lysed in RIPA buffer (Thermo Fisher, catalog #89900), mixed with SDS-loading buffer and denatured at

95 °C for 5 min. The denatured lysates were separated by 12% SDS-PAGE and transferred onto nitrocellulose membranes. The membranes were incubated separately with primary antibodies: anti-HLA-A (1:1000 dilution; ThermoFisher, PA5-29911), anti-HLA-B (1:3000 dilution; abcam, ab193415), anti-HLA-C (1:3000 dilution; abcam, ab193432), anti-HLA-E (1:1000 dilution; abcam, ab300553), anti-IFITM1 (1:1000 dilution; ThermoFisher, MA5-35972), anti-IFITM2 (1:1000 dilution; ThermoFisher, MA5-27503), anti-IL32 (1:500 dilution; ThermoFisher, PA5-119847) and anti-actin (1:5000 dilution; BD Biosciences, 612657). After incubation with primary antibody, the membranes were incubated with secondary antibodies: either goat anti-mouse IgG-HRP (1:10000 dilution; Thermo Fisher, 31430) or donkey anti-rabbit IgG-HRP (1:5000 dilution) (Thermo Fisher, A16023), depending on the source of the primary antibody. Validation statements of all antibodies used for Western blotting were noted on the manufacturer's website with relevant citations. The antibodies were stripped out with strip buffer between each protein detection. Proteins were visualized and imaged with the WesternBright™ Quantum HRP substrate (Advansta, K-12042-D10) and the VILBER FUSION imaging system or the ImageQuant LAS 4000 mini system.

### Enzyme-linked immunosorbent assay

The concentration of C1q in the sera of patients and healthy individuals was quantified using the C1q Human ELISA Kit (Thermo Fisher, BMS2099) according to the manufacturer's instructions.

### siRNA transfection and real-time quantitative polymerase chain reaction (PCR)

My-La CD4+ cells (1,000,000 cells) were transfected with *IL32* siRNA (100 pmol; siRNA ID:140003; Thermo Fisher, catalog #AM16708) or negative-control siRNA (100 pmol; Thermo Fisher, catalog #4390843) using the Neon transfection system (pulse voltage: 1350 V; pulse width: 10 ms; pulse number: 3). Twenty-four hours after transfection, total RNA was extracted using RNeasy Mini Kits (QIAGEN, 74104). The quality and quantity of RNA was measured using a NanoDrop spectrophotometer. cDNA was synthesized from RNA using the SuperScript™ IV First-Strand Synthesis System (Thermo Fisher, catalog #18091050) according to the manufacturer's instructions. The Fast SYBR™ Green Master Mix (Thermo Fisher, catalog #4385612) was applied using the QuantStudio™ 12 K Flex Real-Time PCR System (Thermo Fisher, catalog #4471087). The genes of interest in the real-time quantitative reverse transcription PCR analysis were the *IL32*, *HLA-A*, *HLA-B* and *HLA-C* genes; gene expression levels were quantified using a comparative Ct method and normalized to the expression of the housekeeping gene ActB. The sequences of primers are listed in Supplementary Table 4.

### Complement-dependent cytotoxicity assay

Ramos cells (10,000 cells/well) were washed and plated in a 96-well U-bottom plate in RPMI 1640 medium to serve as targets. Targets were coated with anti-CD20-mAb and opsonized at 37 °C in 5% CO$_2$ for 15 min. Following opsonization, target cells were exposed to serum (10 µl/well) from patients with MF, or L-CTCL, or from healthy individuals. Plates were incubated for 4 h and analyzed for CDC activity using the aCella™ TOX Kit (Cell Technology Inc., catalog #CLA-TOX100-3) or the CyQUANT™ LDH Cytotoxicity Assay Kit (Thermo Fisher, catalog #88954) according to the manufacturer's instructions.

### Antibody-dependent cellular cytotoxicity assay

Isolated NK cells were cultured and rested overnight in complete RPMI 1640 medium supplemented with 100 IU/ml IL2. T cells (10,000 cells/well) that had been washed and plated in a 96-well U-bottom plate in complete RPMI 1640 medium served as targets. Targets were coated with anti-CD52-mAb or anti-CCR4-mAb, and then opsonized at 37 °C in 5% CO$_2$ for 15 min. Following opsonization, target cells were exposed to effector cells in the presence or absence of human anti-MHC-I and

anti-KIR–blocking antibodies. NK cells (effectors) were washed, counted and plated onto targets (100,000 NK cells/well; effector:target ratio, 10:1). Plates were incubated for 24 h, after which ADCC activity was measured.

For live-cell immunofluorescence and ADCC visualization, targets were labeled with Far Red whole-cell stain dye (Thermo Fisher, catalog #C34564), and effector NK cells were labeled with eBioscience™ CFSE (Thermo Fisher, catalog #65-0850-84). Target cells were subsequently exposed to anti-CD52-mAb at a concentration of 10 µg/ml and opsonized at 37 °C in 5% CO$_2$ for 15 min. MHC-I-blocking F(ab')$_2$ fragments were prepared by pepsin digestion and protein A purification and verified by nonreduction SDS-PAGE (Supplementary Fig. 8f). Following opsonization, target cells were exposed to effector cells in the presence or absence of human anti-MHC-I-blocking antibody and anti-MHC-I F(ab')$_2$ fragments for 24 h at 37 °C. Fluorescent images were obtained using the Cytation™ 3 cell imaging multi-mode reader (Bio-Tek Instruments).

For the in vitro mouse ADCC assay, mouse NK cells were isolated from splenocytes using NK Cell Isolation Kit, Mouse (Miltenyi Biotec, catalog #130-115-818) and cultured in complete RPMI 1640 medium. EL4-hCD20 cells were washed and plated in a 96-well U-bottom plate in complete RPMI 1640 medium to serve as targets. Targets were coated with anti-hCD20-mIgG2a and opsonized at 37 °C in 5% CO$_2$ for 15 min. NK cells (effectors) were washed, counted and blocked with anti-Ly49C/I F(ab')$_2$ fragments. Following opsonization, target cells were exposed to effector cells for 4 h at 37 °C.

ADCC activity was analyzed by LDH release (measured using the CyQUANT™ LDH Cytotoxicity Assay Kit [Thermo Fisher, catalog #88954] according to the manufacturer's instructions) or by flow cytometry using propidium iodide (PI) staining. For the PI-FACS assay, targets were first labeled with CFSE, a fluorescent green cell-staining dye, in order to separate them from effectors on flow cytometry. After opsonization with therapeutic antibody and incubation with effectors, PI solution (Miltenyi Biotec, catalog #130-093-233) was added, and ADCC activity was analyzed on Becton Dickinson FACSCanto™ instruments. Data were analyzed using FCS Express 7 Flow Cytometry RUO and Prism V9.1.0 (GraphPad) software.

### Tissue microarray and MACSima imaging platform

Skin biopsies from patients with MF and discarded tissue from surgical margins, serving as healthy controls, were processed into formalin-fixed paraffin-embedded (FFPE) histology blocks using standard pathology procedures. Informed consent was obtained from all patients. A tissue microarray (TMA) was then constructed from these FFPE histology blocks using the TMA Grand Master system (3DHISTECH). Immunostaining was carried out using the MACSima Imaging Platform, with the identification and quantification of CD57+ NK cells achieved through the use of an anti-human CD57 antibody (dilution: 1:50; clone: TB03; Miltenyi Biotec) and 4′,6-diamidino-2-phenylindole (DAPI).

### In vivo mouse tumor model

Wild-type C57BL/6 mice were purchased from Envigo. Age-matched (6–12 weeks) female animals were used throughout experiments. To prevent the potential factors affecting the tumor growth due to sex differences, we used only the female animals in this study. Animal experiments were approved by the Swiss regulatory authorities (license ZH215/2017, 'Anti-cancer therapies based on RNA' and license ZH175/2020, 'Testing and optimizing anti-cutaneous T cell lymphoma immunological treatments in mice') and all mice were kept in accordance with regulations from the Laboratory Animal Services Center at the University Hospital of Zürich. Mice were inoculated subcutaneously with $1 \times 10^5$ EL4-hCD20 tumor cells in 200 µl PBS. Post tumor inoculation, mice received 250 µg of anti-hCD20-mIgG2a or isotype IgG2a control on day 3 and weekly for three weeks, 200 µg of anti-Ly49 C/I F(ab')2 or isotype IgG2a F(ab')$_2$ on day 3 and biweekly for

three weeks or a combination of anti-hCD20-mIgG2a and anti-Ly49 C/I F(ab')₂. Ten mice were used for each experimental group. The therapeutic and blocking antibodies were administered i.p. On day 3 before treatment and on day 9 after treatment, luciferin (D-Luciferin free acid; Synchem UG & Co. KG, catalog #S039) was i.p. injected and an IVIS was used to follow tumor development. Tumor sizes were measured with a caliper three times a week for tumor volume calculation using the equation (width$^2$ × length)/2.

The maximal tumor size permitted by the Swiss regulatory authorities was 1 cm$^3$ and the maximal tumor size was not exceeded in this study. Mice were monitored for tumor volume (cm$^3$), and mice with tumors reaching 1 cm$^3$ were euthanized. To calculate overall survival, mice bearing tumors reaching 1 cm$^3$ and those with open tumors were euthanized.

## RNA extraction and NanoString analysis

Skin biopsies from patients were immediately frozen in liquid nitrogen and stored at −80 °C until processing. RNA was isolated using the TRIzol/chloroform method and a tissue homogenizer (Thermo Fisher Scientific). The quality and quantity of RNA was measured using a NanoDrop spectrophotometer and RNA integrity was analyzed on a Fragment Analyzer (Agilent). mRNA expression was analyzed with the nCounter CAR-T characterization panel and the nCounter Human Immunology V2 panel on the nCounter platform (NanoString Technologies, Seattle, WA, USA) using 100 ng of RNA per skin sample. A quality check was done for each sample and data were normalized and analyzed using nSolver 4.0 (NanoString Technologies). The expression of selected genes was used to generate the heatmap using R. The full NanoString dataset is provided in the Source data file.

## Recombinant human IL-32β labeling and the expression of IL-32β binding sites

Recombinant human IL-32β (rIL-32β) protein (R&D Systems, 6769-IL) was prepared at a concentration of 25 μg/mL (25 μg in 1 mL of PBS) and labeled with APC using the APC Conjugation Kit (Abcam, ab201807) following the manufacturer's instructions.

For flow cytometric analysis, cells were initially stained with fluorescent monoclonal antibodies. After washing, cells were resuspended in an APC-labeled rIL-32β solution at a concentration of 20.8 μg/mL and incubated at 4 °C for 3 h. Subsequently, they were washed with MACS buffer. The stained cells were then acquired (20,000 lymphocytes per sample) using LSRFortessa™ instruments, and the data were analyzed with FCS Express 7 Flow Cytometry RUO (De Novo Software) and Prism v9.1.0 (GraphPad) software.

## IL-32β stimulation analysis

PBMCs and skin T cells were seeded into 48-well plates and stimulated with recombinant human IL-32β (100 ng/mL; R&D Systems, 6769-IL) along with a cocktail of phorbol myristate acetate (PMA) (50 ng/mL), ionomycin (750 ng/mL), and phytohemagglutinin (PHA) (1%) for 48 h. Subsequently, IL-32β-stimulated skin T cells, healthy T cells, and untreated control samples were acquired using LSRFortessa™ instruments, and data were analyzed using FCS Express 7 Flow Cytometry RUO (De Novo Software) and Prism v9.1.0 (GraphPad) software.

## Statistical analysis

GraphPad Prism software (v9.1.0) was used to prepare graphs and perform statistical analysis. Data are shown as individual data points and as mean ± standard error of mean (SEM). Comparison between groups was done through an unpaired, two-tailed Student's $t$-test. Comparison within group was done through a paired, two-tailed Student's $t$-test. Survival curves were plotted by the Kaplan–Meier method and compared using the log-rank test.

## Reporting summary

Further information on research design is available in the Nature Portfolio Reporting Summary linked to this article.

## Data availability

The single-cell RNA seq data of patients with MF generated in this study have been deposited in the GEO database under accession code GSE224449. The single-cell RNA seq publicly available data of healthy controls, MF patients, AD patients, BCC patients and CBCL patients used in this study are available in the GEO database under accession code GSE173205, GSE165623, GSE222840, GSE181907 and GSE173820. The remaining data are available within the Article, Supplementary Information or Source Data file. Source data are provided with this paper.

## Code availability

The neural network logistic regression (NN-log-reg) model is available online with publication at https://github.com/suskim/NN-log-reg[72]. The single-cell RNA seq analysis is available online with publication at https://github.com/GuenovaLab/TumorLymphocytes[73].

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

## Acknowledgements

We would like to thank the University Research Priority Program (URPP) in translational cancer research at the University of Zürich for support of this project. The bioinformatics work on TCR reconstruction was performed by Genevia Technologies. We would like to thank Prof. Jeanette Leusen for providing EL4-hCD20 cell line. We would like to thank Prof. Günter Klambauer for providing the expertise on machine learning. EG, FK, LEF and MPL are members of the SKINTEGRITY.CH collaborative research Program. We thank the editorial support provided by Jo Fetterman and Julie Smith from Parexel International. This work was supported by the Promedica Stiftung (1406/M and 1412/M to EG), the Swiss Cancer Research Foundation (KFS-4243-08-2017 to EG), the Swiss National Science Foundation (IZLIZ3_200253/1 to EG), the European Academy of Dermatology and Venereology (PPRC-2019-20 to EG), the University of Lausanne (SKINTEGRITY.CH collaborative research program to EG), the Fondation Recherche Cancer ISREC (CCP 10-3224-9 to EG) and the Forschungskredit of the University of Zürich (FK-15-040 to WH and FK-17-023 to YTC).

## Author contributions

E.G. and W.H. designed the project and had the oversight. E.G. and Y.T.C. conceived and designed the experiments. Y.T.C, F.K., P.M.B. and D.I. acquired and flow cytometry and single-cell RNA seq data. S.K. performed the artificial neural network machine learning. Y.T.C and P.P. performed computational data analysis. C.I. prepared the tissue micro-array. Y.T.C. and M.B. performed in vitro ADCC assays. Y.T.C. performed, S.P. and E.G. advised the mouse experiments. Y.T.C., Y.C.T., O.P. and E.G. prepared figures. E.G., M.P.L., L.E.F. and F.K. collected clinical samples and clinical metadata. Y.T.C., D.I., S.K. and E.G. wrote the materials and methods. Y.T.C., S.K. and E.G. drafted the results. E.G. and W.H. conceived and supervised the project. Y.T.C, Y.C.T. and E.G. sharped the final version of the manuscript. All authors approved the final version of the manuscript.

## Competing interests

The authors declare no competing interests.

## Additional information

[1]Department of Dermatology, Lausanne University Hospital (CHUV) and Faculty of Biology and Medicine, University of Lausanne, Lausanne, Switzerland. [2]Department of Dermatology and Venerology, Medical Faculty, Johannes Kepler University, Linz, Austria. [3]Department of Dermatology, University Hospital of Zurich and Faculty of Medicine, University of Zurich, Zurich, Switzerland. [4]Department of Dermatology and Allergology, Ludwig-Maximilians-University of Munich, Munich, Germany. [5]Dr. Phillip Frost Department of Dermatology and Cutaneous Surgery, University of Miami Miller School of Medicine, Miami, FL, USA. [6]Department of Immunology, Medical University of Warsaw, Warsaw, Poland. [7]Department of Dermatology, Icahn School of Medicine at Mount Sinai, New York, USA. [8]Department of Dermatology, Hospital 12 de Octubre, Medical School, University Complutense, Madrid, Spain. [9]These authors jointly supervised this work: Wolfram Hoetzenecker, Emmanuella Guenova. ✉e-mail: wolfram.hoetzenecker@kepleruniklinikum.at; emmanuella.guenova@unil.ch

