## [Peer Review File · Nature Communications]

MHC-I Upregulation Safeguards Neoplastic T Cells in the Skin Against NK Cell-mediated Eradication in Mycosis FungoidesREVIEWERS' COMMENTS:

Reviewer #1 (Remarks to the Author): with expertise in skin cancer, immunology, omics

In this manuscript, the authors pursue single cell RNA-Seq to understand the biology of cutaneous T cell lymphoma. Utilizing a machine learning model, they identify MHC Class I as a gene that is a classifier of tumor cells in the skin but not bystander cells, healthy cells from uninvolved skin, or the blood. They go on through the functional assays to suggest that the increased MHC class I molecule expression may mediate resistance to Nk cell-mediated ADCC.

The manuscript is provocative. However, for robustness and for comparison to contemporary literature, additional experiments/explanation are required.

Major:

1. The number of cells isolated appear very low. the authors report only 1174 tumor cells from skin and 573 cells from blood. This makes it difficult to understand the robustness of their data. While isolating T cells from skin is non-trivial, the number of cells isolated from blood from presumably Sezary patients appears extremely low.
2. Related to #1, it is not clear why this machine learning algorithm is needed. Perhaps, the authors need to be clearer with their data. However, they already have a method (TCR-sequencing) to distinguish clonal tumor cells from other cells in the microenvironment. Why do they need this machine learning approach?
3. The authors should explain how or why their method is better than others. If I were to hazard a guess, I would think this would be the only way they could analyze such a small number of cells. Perhaps not all of them had TCR-sequencing so they had to extrapolate from signatures to cells wherein TCR sequences could not be derived?
4. The authors should try to analyze their data across traditional means. It is not clear if and how this method performs better than other analysis tools, such as Seurat. If it does better,

this should be explained.

5. Returning to point 1, the number of cells for the functional assays (Western blots, flow Cytometry, and NK cell assays) are not clear. If the group can isolate these cells with higher numbers, they should try to isolate higher numbers of single cell data for statistical robustness. 1173 cells across 17 tumors is low and may be subject to multiple sources of bias.

6. For the functional data, the primary data for the NK cell assays should be provided. if they can only isolate ~80 cells per tumor, it should be clear how they interpreted their results. if they get a higher number, they should try to explain why there is a huge discrepancy with the single cell in number of samples.

7. The IL-32 data is provocative but underdeveloped. Simple assays from biochemistry to transcriptomics could demonstrate whether IL-32 is sufficient to induce this phenotype in normal cells and provide some mechanism.

8. With their single-cell RNA-seq, the authors should provide a putative mechanism by which IL-32 is unregulated in tumor cells.

9. What is the relative number of MHC Class I molecules in the Sezary samples, the MF samples, and the mouse model of EL-4?

Minor:

1. I found the title misleading. I think their data suggests mycosis fungoides specifically. perhaps it should be in the title.

2. the description of samples and QC of sample prep should be more clearly laid out for single cell and subsequently for the functional data considering the low number of cells.

Reviewer #2 (Remarks to the Author): with expertise in skin cancer, immunology, omics

The paper of Chang et al. proposes a mechanism for the resistance of malignant T-cells in the skin of patients with mycosis fungoides to the therapeutic antibodies. The authors believe that IL32-driven expression of HLA I inhibits the interaction with the ADCC-mediating NK cells. Inhibition of interaction increases tumor eradication in a mouse model and increased ADCC in an in vitro assay.

The findings are original, interesting, and may have therapeutic consequences in treating cutaneous malignancies. The experimental plan is well designed, starting from the sequencing data followed by the experimental verification in vitro and in mouse models. However, some of the experimental details and analyses are not clearly described, which precludes enthusiastic acceptance of the authors' findings at this stage of submission.

1. One of my most important concerns is the sample size.

a. The authors managed to sequence a total of 1,174 malignant T-cells obtained from 17 lesions from 14 patients. This is only 69 cells/sample on average. Considering that most scRNAseq experiments are done on 5,000-10,000 cells, the number of cells seems very small and I would question whether the sample was representative. A formal sample size assessment would be necessary to assess the probability of bias/error. Also, I cannot find the number of cells for each sample (hopefully, the dataset is not dominated by the cells from a few, selected samples).

b. I am even more concerned regarding the validity of AI training done on 1000+ cells. What is the risk of data overfitting and what control experiments were done to ascertain that the results are not overfitted? What was the sample size for the training and validation datasets? For this part of the study, I would like to see a review from a researcher who is a specialist in AI.

2. Figure 1.

a. Fig 1 B has 13 samples - what happened to the remaining samples?

b. Clonality rules. I have to admit that I do not understand this part of the methods. It is mentioned that the TCR alpha-beta pairing was determined by choosing the most abundant alpha and beta clonotypes. However, extended Figure 1 shows that this was not always the

case. Assuming, that the colored clonotypes are those from cancer cells (this is an assumption because legend does not say what the coloring means) those are not always the most abundant. Some samples (e.g. MF029, MF056) show several expanded clonotypes. How do we know which of those are malignant and which are bystander T cells? Also, for the true cell clones, the frequency of TCR beta should be the same as the alpha (or at least comparable). This is not the case, e.g. in MF057 the dominant TRBV18 is paired with TRAV6 rather than the more abundant TRAV9_2.

c. Fig 1c: It is not clear how this was done. More explanation to this figure should be provided.

3. Figure 2 - Pseudotime progression. I have again a question regarding the validity of data based on very few cells. The progression data was done in 3 patients and I wonder what is the number of cells used for analysis. How is this representative?

4. Figure 3.

a. Fig 3 d shows only a few NK cells in the skin. It seems to me that this is the best explanation for the resistance of skin cells to ADCC in vitro. Although the resistance to ADCC shown later in vitro (at the 10x excess of NK cells over the target) may be a real thing, just a mere lack of NK cells in skin infiltrate is a much more straightforward explanation.

b. In panel 3f - the decrease in NK numbers seems to quite small. What is the proof that this decrease actually is functionally important?

c. Panel 3g. More experimental detail to explain the cytotoxicity assays is needed. First, I am surprised that NK cells can kill non-activated autologous T-cells. This needs more explanation, as it is believed that resting T-cells are not sensitive to NK-mediated ADCC.

Then, what is the number of independent repeats (different samples from different patients) for the ADCC experiments? I assume that tumor cells were sampled in addition to those used for sequencing experiments. Which patients in which stages were included here? Which antibody was used? CD52, CCR4 or both?

5. Figure 6. IL32 experiments. To my knowledge, IL32 is very commonly expressed in any inflammatory skin disease (and also in inflamed joints) and can be expressed by macrophages, T-cells or even fibroblasts. This is by no means specific to CTCL. ^[11]The

involvement of IL32 is based on correlation and mechanistic experiments (e.g. siRNA downregulation) would be needed to make a link between IL32, HLA I expression, and sensitivity to ADCC.

Reviewer #3 (Remarks to the Author): with expertise in cancer immunology, NK cell biology

In this manuscript, the authors perform ScRNA-seq on tumor cells from skin derived T cell mycosis fungoides. They find that NK cell function is down, and this may be explained by HLA-A upregulation mediated by IL-32 and that blockade (of HLA or KIR) can restore function and antitumor activity.

One of the biggest concerns I have with the manuscript is based on the NK cell biology theme. The authors interchangeably talk about MHC class I and KIR interactions but the data they have is only for HLA-A. Oddly, HLA-A, except for perhaps a few alleles, is not generally an inhibitory ligand for killer immunoglobulin-like receptors (KIR) making this story rather confusing. KIR2DL1/L2/L3 binds HLA-C (group 1 and group 2 ligands) and KIR3DL1 binds HLA-Bw4. The data in figure 1 shows HLA-A but also HLA-E, a ligand for NKG2A. Figure 2 only shows HLA-A, and the KIR ligands for HLA-B and HLA-C need to be shown to make any sense of how this is presented. Perhaps HLA-E is the dominant mechanism and monalizumab would be effective as well.

In figure 1g, HLA-I staining is shown. This is likely also picking up HLA-E? For extended data 4a, what is the MFI of the negative control? The correlation between HLA expression and function needs to be more clear.

The NK cells in 3F are incredibly low even in healthy skin making the physiologic relevance of the differences hard to ascertain.

The enhanced function with anti-KIR in figures 4e/f is modest, raising the questions above.

There is confusion jumping from human to mouse. Mice do not have KIR and greater care in

some of the transitions is needed. Even the abstract inaccurately talks about KIR blockade in mice in vivo as well as lines 263-4. I am not sure that the mouse data is completely relevant to the story and translatable given that mice do not have KIR.

Because of all these issues, the last paragraph of the discussion seems speculative and conclusions do not seem definitive.

POINT-BY-POINT RESPONSE TO ALL COMMENTS RAISED BY REVIEWER 1:

Reviewer #1 (Remarks to the Author): with expertise in skin cancer, immunology, omics

General Comment 1: In this manuscript, the authors pursue single cell RNA-Seq to understand the biology of cutaneous T cell lymphoma. Utilizing a machine learning model, they identify MHC Class I as a gene that is a classifier of tumor cells in the skin but not bystander cells, healthy cells from uninvolved skin, or the blood. They go on through the functional assays to suggest that the increased MHC class I molecule expression may mediate resistance to Nk cell-mediated ADCC.

The manuscript is provocative. However, for robustness and for comparison to contemporary literature, additional experiments/explanation are required.

General Answer 1: Your feedback is greatly appreciated. We're glad to hear that you find our manuscript thought-provoking. In response to your valuable input, we've made extensive revisions that include additional experiments, refined wording, and detailed explanations. We hope that the updated version now fully meets your expectations.

Major:

Comment 1. 1: The number of cells isolated appear very low. the authors report only 1174 tumor cells from skin and 573 cells from blood. This makes it difficult to understand the robustness of their data. While isolating T cells from skin is non-trivial, the number of cells isolated from blood from presumably Sezary patients appears extremely low.

Answer 1.1: Thank you sincerely for raising this important point. Your feedback has illuminated areas in our methods that needed further clarity.

To answer shortly, the lower number of cells compared to other single-cell RNA study is due to the choice of the technology, not the difficulty in isolating the cells. However, the lower number of cells is compensated by 1) high number of sequenced genes per cell 2) high coverage of these genes 3) good quality TCR reconstruction (number of genes per cell = 40, 299; average of number of reads per cell = 888, 317; percentage of recovered TCR = 71.8%)

In more details:

We employed the Fluidigm C1 Single-Cell platform (Smart-seq2) for scRNA-seq, a method with enhanced sensitivity and quality for TCR reconstruction. However, it's essential to note that this method has a limited capacity to capture and process a maximum of 96 individual cells per sample.

The Fluidigm C1 platform uses microfluidics to isolate single cells in individual reaction chambers, where cell lysis, reverse transcription, and cDNA amplification occur. A major advantage of the Fluidigm C1 platform is that it allows for detailed analysis of individual cells with high sensitivity and quality, making it particularly suitable for applications where depth of analysis is a priority. When it comes to TCR sequencing, the Fluidigm C1 platform readily captures and sequences TCR information from individual cells. As a disadvantage, its lower throughput compared to, for example, the 10X platform results in fewer cells being sequenced per sample.

The 10X single-cell RNA platform is a popular technique used for single-cell RNA sequencing. Compared to the Fluidigm C1 platform, it substantially differs in terms of its methodology and capability, particularly when it comes to TCR (T-cell receptor) sequencing. The 10X Genomics platform employs droplet-based technology to encapsulate individual cells in droplets along with barcoded beads that capture and label the RNA from each cell. As an advantage, the 10X platform allows for high-throughput analysis, meaning it can process many cells simultaneously. This results in a much higher number of cells sequenced per sample compared to the Fluidigm C1 platform. However, the 10X platform is designed to capture RNA transcripts, and while it can capture TCR information, it might not offer the same level of specificity and resolution for TCR sequencing as the Fluidigm C1 platform. For further insight, a comprehensive comparison of various single-cell RNA-sequencing methods can be found in the publication

by Ding et al. titled "Systematic comparison of single-cell and single-nucleus RNA-sequencing methods" (Ding, J. et al. *Nat Biotechnol* 38, 737–746, 2020 (<https://doi.org/10.1038/s41587-020-0465-8>)).

It's important to note that advancements in sequencing technologies continue to evolve, so the capabilities of these platforms change over time and new platforms constantly develop. For our experiments and at the time point of their execution, the Fluidigm C1 platform best aligned with our study goals. With the Fluidigm C1 platform we achieved a remarkable single skin T cell capture rate of 87.2% and a high TCR β chains reconstruction rate of 71.8%.

We have provided the reasoning for the selection of the Fluidigm C1 platform as well as an explicit description of its advantages and limitations in the "Results" section (page 5, line 123-126). The data from the additionally incorporated large-scale single-cell RNA dataset [GSE173205] generated through the 10X single-cell RNA sequencing are presented in the new Extended Data Fig. 4A. We hope this additional information addresses your concerns.

Comment 1.2: Related to #1, it is not clear why this machine learning algorithm is needed. Perhaps, the authors need to be clearer with their data. However, they already have a method (TCR-sequencing) to distinguish clonal tumor cells from other cells in the microenvironment. Why do they need this machine learning approach?

Answer 1.2: Thank you for your valuable feedback, and we apologize for any ambiguity in our explanations.

In the context of MF skin lesions, differentiating malignant T cells from benign bystander skin T cells presents challenges due to:

- *Molecular and phenotypic heterogeneity in CTCL, lacking distinct markers.*
- *Often low abundance of malignant T cells amid healthy ones.*
- *Morphological and functional similarity between tumor and healthy T cells.*
- *High genetic variability among patients.*

*While TCR monoclonality can be a marker, it doesn't capture transcriptomic differences. Machine learning methods are suitable for analyzing the gene expression dataset that has a limited number of samples but a large number of features (Ravindran, U. et al. *Prog Biophys Mol Biol* 177, 1-13, 2023 (<https://doi.org/10.1016/j.pbiomolbio.2022.08.004>)).*

As we sequenced deeply only on T cells with small number of cells but high coverage of sequenced genes, machine learning methods better adapted to find features distinguishing two T-cell subpopulations (tumor and non-tumor T cells). Moreover, the ANN approach captures all important genes, whether they are upregulated or downregulated. This unbiased analysis enabled exploration of complex transcriptome patterns via machine learning, ideal for evolving high-dimensional data with nonlinear relationships. Subsequent traditional statistical and functional analyses (Figures 2-6) followed, enhancing our understanding of tumor T cells in CTCL.

Comment 1.3: The authors should explain how or why their method is better than others. If I were to hazard a guess, I would think this would be the only way they could analyze such a small number of cells. Perhaps not all of them had TCR-sequencing so they had to extrapolate from signatures to cells wherein TCR sequences could not be derived?

Answer 1.3: We value your insightful comments and appreciate the opportunity to address them. It is important to clarify that we did not extrapolate from signatures to T cells without TCR reconstruction. Our emphasis was exclusively on TCR-reconstructed T cells, ensuring the analysis's robustness and reliability. We have added the text to explain more clearly in the "Methods" section (page 24, line 634-635).

Given the intricate tumor T cell heterogeneity in CTCL, we believe that our method's strict TCR-reconstructed focus enabled insights that other approaches might miss. This adequately addresses the challenges posed by CTCL's heterogeneity and the small cell count with high coverage of sequenced genes. This rigorous methodology allowed us to guarantee best possible with current methods sensitivity,

quality, and dependability in our analysis, fostering a comprehensive grasp of tumor T cells within this context.

Comment 1.4: The authors should try to analyze their data across traditional means. It is not clear if and how this method performs better than other analysis tools, such as Seurat. If it does better, this should be explained.

Answer 1.4: Seurat is an effective tool for analyzing large numbers of cells with distinct gene signatures in contexts like 10X single-cell RNA-seq for PBMCs. However, Seurat encounters challenges in differentiating highly similar cell types (Huang, Q. et al. Genomics, Proteomics & Bioinformatics 19, 267-281. 2021 (<https://doi.org/10.1016/j.gpb.2020.07.004>)) and detailed cell subtype analysis (Liu, J. et al. Front. Genet 12, 655536. 2021 (<https://doi.org/10.3389/fgene.2021.655536>)). Given our focus exclusively on T cells and the nuanced analysis of T-cell subtypes (tumor and non-tumor) in this study, the implementation of a machine learning algorithm for tumor-related genes offers greater accuracy.

We appreciate your suggestion and subsequently conducted Seurat analysis with our single-cell data. To make sure the results we obtained through the machine learning approach are robust across methodology or dataset we have:

- Conducted a Seurat analysis with our single-cell data. This analysis revealed significant upregulation of classical HLA-I genes in certain tumor-T-cell clusters, compared to healthy- and bystander-T-cell clusters. These findings align with our attention-ANN results, further reinforcing the credibility of our conclusions. To avoid redundancy, at this stage we considered it better not to include the plots generated by our Seurat analysis in the revised manuscript, but will be happy to do so if you and the Editor deem it necessary.*
- Re-analyzed a large-scale single-cell RNA dataset [GSE173205] generated through the 10X single-cell RNA sequencing that covers patch (early stage) and plaque/tumor (late stage) MF skin lesions (new Extended Data Fig. 4a). This alternative approach further supports and enhances our observations related to the expression of classical MHC-I genes—HLA-A, HLA-B, and HLA-C—in MF tumor T cells from plaque/tumor late-stage skin lesions.*

Comment 1.5: Returning to point 1, the number of cells for the functional assays (Western blots, flow Cytometry, and NK cell assays) are not clear. If the group can isolate these cells with higher numbers, they should try to isolate higher numbers of single cell data for statistical robustness. 1173 cells across 17 tumors is low and may be subject to multiple sources of bias.

Answer 1.5: We apologize once again for any confusion arising from the lack of clarity in our presentation of cell numbers isolated from the skin for various analyses. As already outlined in our Response to your Comment 1.1, we chose the Fluidigm C1 platform for TCR sequencing and simultaneous transcriptome analysis at the single-cell level to capitalize on the platform's advanced capabilities for TCR analysis, albeit with a trade-off in terms of throughput.

For Western blot analysis, we used 1,000,000 cells. For flow cytometry, we acquired at least 10,000 cells per sample. For NK cell assays, we used 10,000 T cells as target cells and 100,000 NK cells as effectors. The related part of the Methods has been updated accordingly (Western blot: page 25, line 684; Flow Cytometry: page 23, line 601; NK-cell assays: page 27, line 730-737).

Comment 1.6: For the functional data, the primary data for the NK cell assays should be provided. if they can only isolate ~80 cells per tumor, it should be clear how they interpreted their results. if they get a higher number, they should try to explain why there is a huge discrepancy with the single cell in number of samples.

Answer 1.6: Regarding the total cell numbers used in the different experiments, please see our Response to your Comments 1-5 above.

For your convenience, the primary data from the NK cell assays presented in Figures 3g-h, 4a, 4c, 4e-f and 5b are accessible in the Supplementary Raw Data Repository Files under the respective Excel files Fig3g_ADCC in blood and skin.xlsx, Fig3h_ADCC in skin of MF.xlsx, Fig4a_ADCC HLA blocked.xlsx,

Fig4c_ADCC based on images.xlsx, Fig4e_FACS_ADCC_KIR blockade.xlsx, Fig4f_LDH_ADCC_KIR blockade, Fig5b_FACS_ADCC.xlsx and Fig5b_LDH_ADCC.xlsx. To address the specifics of the NK cell assays, we used 10,000 T cells as target cells and 100,000 NK cells as effectors, as outlined in the Methods section on page 27, line 730-737.

Comment 1.7: The IL-32 data is provocative but underdeveloped. Simple assays from biochemistry to transcriptomics could demonstrate whether IL-32 is sufficient to induce this phenotype in normal cells and provide some mechanism.

Answer 1.7: We're pleased to hear that you share our enthusiasm for IL-32's unexpected role in MF. Your insightful feedback is greatly appreciated and has motivated us to delve further into this aspect of our research. Following your valuable suggestion, we performed additional experiments to enrich our analysis of IL-32 within the CTCL context. Notably, we discover that tumor T cells originating from MF skin lesions exhibit significantly elevated levels of IL32 binding sites (IL32BC) when compared to their normal T cell counterparts. Furthermore, we demonstrate that the stimulation of IL32 leads to the exclusive upregulation of classical MHC-I (HLA-A, B, C) expression in MF tumor T cells, while leaving normal T cells unaffected. These newly obtained mechanistic insights into the role of IL32 in CTCL are presented in the "Results" (page 15, line 402-414) and in the newly incorporated in the revised manuscript Fig. 6h-i. The related part of the Methods has been updated accordingly (page 30, line 802-819).

Comment 1.8: With their single-cell RNA-seq, the authors should provide a putative mechanism by which IL-32 is unregulated in tumor cells.

Answer 1.8: In response to the valuable suggestion from the reviewer, we have conducted further analysis on genes that positively correlate with IL32 expression in clonal tumor T cells. Our investigation identifies "LGALS1" and "S100A4" as top two genes positively correlated with IL32 in tumor T cells. Based on existing literature, both genes are associated with the inhibition of cell apoptosis. This discovery allows us to formulate a hypothesis that the upregulation of IL32 in tumor T cells might serve to promote the survival of these cells. This hypothesis finds support in previous studies focusing on IL32 in CTCL. Notably, it has been reported that IL32 can effectively enhance the proliferation of tumor T cells (Suga, H. et al. JID. 2014) and extend the survival of malignant T-cell clones (Yu, KK. et al. JID. 2022). Additionally, IL32 has been demonstrated to induce CD14⁺ antigen-presenting cells to provide support for the survival of malignant T cells (Yu, KK. et al. JID. 2022). This convergence of evidence from various publications reinforces our hypothesis and highlights the potential role of IL32 in facilitating the survival and proliferation of MF tumor T cells.

In our revised manuscript, we have thoroughly explored the existing knowledge surrounding IL32 within the context of CTCL, and we have appropriately referenced the abovementioned papers to support our discussion (page 17, line 464-472; reference 42; reference 55; reference 56). However, concerning the top IL32-related genes identified in tumor T cells, we would prefer to withhold this data from the current manuscript. Given the manuscript's already substantial content, we believe that reserving this information for a separate research project, currently being conducted in our laboratory, would be more prudent. This approach will allow us to delve into the role of IL32 and its associated genes in a more focused manner, without further expanding the length of the present manuscript. However, if the reviewer or editor deems it necessary, we are prepared to include the data as a supplementary figure prior to publication.

Comment 1.9: What is the relative number of MHC Class I molecules in the Sezary samples, the MF samples, and the mouse model of EL-4?

Answer 1.9: We can assess the relative expression of MHC Class I through fluorescence intensity measurements using fluorescein-conjugated anti-MHC Class I antibodies via flow cytometry. According to our mean fluorescence intensity (MFI) findings, classical MHC-I's relative average expression in SS clonal blood tumor T cells is approximately 30,000, while in MF clonal skin tumor T cells, it is around 70,000. In mice, the relative average expression of classical MHC-I in normal T cells from the spleen and blood is 4,500. Contrarily, the relative average expression of classical MHC-I in mouse EL4 T-cell tumors is notably higher, reaching 24,695.

This data is presented graphically in Figure 2h and 5a.

Minor:

Comment 1. 10: I found the title misleading. I think their data suggests mycosis fungoides specifically. perhaps it should be in the title.

Answer 1.10: We fully concur. The title has been adjusted to encompass the proposed changes and mirror the revised version of our manuscript.

Comment 1.11: the description of samples and QC of sample prep should be more clearly laid out for single cell and subsequently for the functional data considering the low number of cells.

Answer 1.11: Thank you for your valuable input. We acknowledge the need for clearer delineation of sample descriptions and QC processes, especially given the limited cell numbers involved. In response to your suggestion, we have included comprehensive descriptions of the sample preparation process and quality controls for both single-cell analysis and the constraints associated with using the Fluidigm C1 single-cell RNA-sequencing platform in both the "Results" and "Methods" sections (page 6, line 134-135; page 23, line 615-618).

With this, we believe to have enhanced the clarity and detail of our sample preparation procedures for both single-cell analysis and subsequent functional data assessment.

POINT-BY-POINT RESPONSE TO ALL COMMENTS RAISED BY REVIEWER 2:

Reviewer #2 (Remarks to the Author): with expertise in skin cancer, immunology, omics

General Comment 2: The paper of Chang et al. proposes a mechanism for the resistance of malignant T-cells in the skin of patients with mycosis fungoides to the therapeutic antibodies. The authors believe that IL32-driven expression of HLA I inhibits the interaction with the ADCC-mediating NK cells. Inhibition of interaction increases tumor eradication in a mouse model and increased ADCC in an in vitro assay. The findings are original, interesting, and may have therapeutic consequences in treating cutaneous malignancies. The experimental plan is well designed, starting from the sequencing data followed by the experimental verification in vitro and in mouse models. However, some of the experimental details and analyses are not clearly described, which precludes enthusiastic acceptance of the authors' findings at this stage of submission.

General Answer 2: We sincerely appreciate your acknowledgment of the originality and novelty of our data. Furthermore, we would like to extend our heartfelt gratitude for your valuable and insightful comments.

Comment 2.1.a: One of my most important concerns is the sample size. a) The authors managed to sequence a total of 1,174 malignant T-cells obtained from 17 lesions from 14 patients. This is only 69 cells/sample on average. Considering that most scRNAseq experiments are done on 5,000-10,000 cells, the number of cells seems very small and I would question whether the sample was representative. A formal sample size assessment would be necessary to assess the probability of bias/error. Also, I cannot find the number of cells for each sample (hopefully, the dataset is not dominated by the cells from a few, selected samples).

Answer 2.1.a: Thank you for your comment, which has also been raised by Reviewer 1 (please also see our Answer 1.1).

In response to your valuable feedback (Answer 1.1), we appreciate the opportunity to clarify our methodology. The lower number of cells in our single-cell RNA study is primarily a result of our choice of technology, rather than difficulties in cell isolation. However, the lower number of cells is compensated by 1) high number of sequenced genes per cell 2) high coverage of these genes 3) good quality TCR

reconstruction (number of genes per cell = 40, 299; average of number of reads per cell = 888, 317; percentage of recovered TCR = 71.8%)

Specifically, we utilized the Fluidigm C1 Single-Cell platform (Smart-seq2), a method known for its sensitivity and TCR reconstruction quality. However, it has a limited capacity to process a maximum of 96 individual cells per sample due to its microfluidics-based isolation and amplification process.

Comparatively, the 10X Genomics platform offers higher throughput but may have limitations in TCR sequencing specificity and resolution. The choice of platform was aligned with our study goals, with the Fluidigm C1 platform providing strong single skin T cell capture rate of 87.2% and a TCR β chains reconstruction rate of 71.8%.

We have detailed the advantages and limitations of our chosen platform in our "Results" section. Additionally, we have incorporated data from a large-scale single-cell RNA dataset generated using the 10X platform to complement our study.

Regarding your question about the exact number of cells for each sample – this data is comprehensively provided in Supplementary Table 2. We can affirm that there was no bias due to an overrepresentation of cells from specific samples.

We hope this information addresses your concerns and provides clarity on our methodology.

Comment 2.1.b: I am even more concerned regarding the validity of AI training done on 1000+ cells. What is the risk of data overfitting and what control experiments were done to ascertain that the results are not overfitted? What was the sample size for the training and validation datasets? For this part of the study, I would like to see a review from a researcher who is a specialist in AI.

Answer 2.1.b: We acknowledge your concern regarding the AI training on a dataset comprising over 1000 cells. Despite the relatively small number of cells, it's important to note that these are single T cells, sequenced with high sensitivity and quality, resulting in a substantial number of detected genes (n=44,782). We believe this dataset is well-suited for our machine learning approach.

In terms of the sample size breakdown, our training dataset included 485 clonal malignant skin T cells and 88 non-clonal bystander T cells. For validation, we used 182 clonal malignant skin T cells and 70 non-clonal bystander T cells. We have taken measures to address the risk of overfitting, including the rigorous partitioning of data into training and validation sets.

Comment 2.2.a on Figure 1: Fig 1 B has 13 samples - what happened to the remaining samples?

Answer 2.2.a on Figure 1: We conducted sequencing on a total of 14 patients. However, one sample (MF040_S) posed a challenge as only 4 cells passed the FastQC control, and none of them had a successfully reconstructed TCR. Consequently, we opted not to include any cells from this sample in our subsequent analyses. Nonetheless, we retained this sample in the manuscript to calculate the malignant clone detection rate using mathematical TCR reconstruction. Hence, 13 samples are presented in Fig 1b.

Comment 2.2.b: Clonality rules. I have to admit that I do not understand this part of the methods. It is mentioned that the TCR alpha-beta pairing was determined by choosing the most abundant alpha and beta clonotypes. However, extended Figure 1 shows that this was not always the case. Assuming, that the colored clonotypes are those from cancer cells (this is an assumption because legend does not say what the coloring means) those are not always the most abundant. Some samples (e.g. MF029, MF056) show several expanded clonotypes. How do we know which of those are malignant and which are bystander T cells? Also, for the true cell clones, the frequency of TCR beta should be the same as the alpha (or at least comparable). This is not the case, e.g. in MF057 the dominant TRBV18 is paired with TRAV6 rather than the more abundant TRAV9_2.

Answer 2.2.b on Figure 1: Thank you for your feedback, and we appreciate your patience in clarifying this aspect of our methods. We understand the confusion surrounding TCR α - β pairing and have taken steps to address it.

To avoid ambiguity, we have removed the previous Extended Data Figure 1 and provided a more detailed explanation in the Extended Data Figure 2, improved for better readability (enlarged images). Our "clonality rules" are as follows:

- *Main-Clone: This group comprises cells with the most abundant combination of α and β chains. For instance, if the majority of cells share the combination “ $\alpha 1$ with $\beta 1$ ”, the “main-clone” cells should have either “ $\alpha 1$ with $\beta 1$ (or $\beta 2$)” or “ $\alpha 1$ (or $\alpha 2$) with $\beta 1$ ”.*
- *Related-to-Main-Clone: These cells do not match the α or β chains of the “main-clone” but have a relative connection with the “main-clone” cells, such as “ $\alpha 2$ with $\beta 3$ (or $\beta 4$)” or “ $\alpha 3$ (or $\alpha 4$) with $\beta 2$ ”.*
- *Bystander Groups: This category includes cells where more than two share the same α or β chains, but they do not have any connection to the α or β chains of the “main-clone”.*
- *Single Bystanders: These are individual cells that do not share α or β chains with any others in the sample.*

After TCR reconstruction, most cells possess both "TCR α -chain" and "TCR β -chain." The "main-clone" is defined based on the largest number of cells sharing the same "combination" of α - and β -chains, rather than merely the most abundant individual α -chain and β -chain.

In cases where samples like MF029 and MF056 have multiple expanded clonotypes, they are classified as "related-to-main-clone" since they maintain some connection to the "main-clone." Cells sharing the same α or β chains but lacking a connection to the "main-clone" are designated as "bystander groups."

Regarding the frequency of TCR β -chain compared to α -chain, while it is ideal for them to be comparable, the construction rate for β -chain (71.8%) significantly surpasses that of α -chain (48.1%). For example, in MF057, even though TRBV18 pairs with TRAV6, the presence of TRAV9-2 in the sample prompted us to classify all these cells as part of the "main-clone."

We hope this detailed explanation clears up any confusion, and we appreciate your help and efforts in reviewing our methods.

Comment 2.2.c to Figure 1: Fig 1c: It is not clear how this was done. More explanation to this figure should be provided.

Answer 2.2.c. to Figure 1: Thanks for the suggestion. To provide further clarity, we have included detailed explanations in the figure legend of the Extended Data Figure 2, improved for better readability (enlarged images).

Comment 2.3: Figure 2 - Pseudotime progression. I have again a question regarding the validity of data based on very few cells. The progression data was done in 3 patients and I wonder what is the number of cells used for analysis. How is this representative?

Answer 2.3: We apologize for not been very clear here. In Figure 2, which depicts pseudotemporal progression, we used Monocle 3 computational tool. The analysis is based on data from 13 patients (the sample from the 14th patient MF040_S was excluding due to the absence of TCR-reconstructed cells). We have now reworded the text to improve the clarity of our message.

Comment 2.4.a on Figure 3: Fig 3 d shows only a few NK cells in the skin. It seems to me that this is the best explanation for the resistance of skin cells to ADCC in vitro. Although the resistance to ADCC shown later in vitro (at the 10x excess of NK cells over the target) may be a real thing, just a mere lack of NK cells in skin infiltrate is a much more straightforward explanation.

Answer 2.4.a on Figure 3: We appreciate your observation and concern regarding the presence of NK cells in the skin, which could explain the observed resistance of skin cells to ADCC in vitro.

It's noteworthy that NK-mediated tumor-cell-surface-antigen targeted therapy, which is effective for some skin disorders like cutaneous B-cell lymphoma (CBCL), may not necessarily correlate with a significant increase in the percentage of skin NK cells in the lesions of CBCL. This suggests that the resistance to treatment in certain skin disorders may have other contributing factors beyond the proportion of skin NK cells.

Notwithstanding, to address your comment, we conducted additional experiments to quantify deeper NK cells in diseased and healthy skin. Our findings, as depicted in the newly added Figure 3d-e,

reveal that CD57+ NK cells are indeed present in both healthy individuals and typical MF skin lesions. Furthermore, the percentage of CD57+ NK cells is significantly elevated in the skin of MF patients compared to healthy individuals. Additionally, we sought to gain insights into the proportion of skin NK cells in various skin disorders and healthy skin by analyzing publicly available single-cell RNA datasets for mycosis fungoides (MF: GSE173205 and GSE165623), atopic dermatitis (AD: GSE222840), basal cell carcinoma (BCC: GSE181907), and cutaneous B-cell lymphoma (CBCL: GSE173820). Using a specific gene signature [NKG7, KLRB1, KLRC1, KLRD1, KLRK1, CD7, GZMB, GNLY, NCAM1, GZMH, CCL4, IFNG, CCL4L2, FCGR3B and FCGR3A], we identified NK cells and found that, on average, they constitute approximately 2% of the immune cell population in the skin. Importantly, there was no significant difference in the percentage of skin NK cells between skin cancers/diseases and healthy skin biopsies, as detailed in the new Extended Data Fig. 6 b-c.

We hope this additional information clarifies the role of NK cells in the context of skin disorders and their response to treatment.

Comment 2.4.b on Figure 3: In panel 3f - the decrease in NK numbers seems to quite small. What is the proof that this decrease actually is functionally important?

Answer 2.4.b on Figure 3: We appreciate your observation regarding the decrease in NK numbers in Figure 3f. We completely agree with you and think that this decrease does not appear to be functionally important. To avoid ambiguity of our message, we have decided to remove the data in question and replace it with our newly obtained results for CD57+ NK cell detection, which we believe provides a more relevant and meaningful insight.

Comment 2.4.c on Figure 3: Panel 3g. More experimental detail to explain the cytotoxicity assays is needed. First, I am surprised that NK cells can kill non-activated autologous T-cells. This needs more explanation, as it is believed that resting T-cells are not sensitive to NK-mediated ADCC. Then, what is the number of independent repeats (different samples from different patients) for the ADCC experiments? I assume that tumor cells were sampled in addition to those used for sequencing experiments. Which patients in which stages were included here? Which antibody was used? CD52, CCR4 or both?

Answer 2.4.c on Figure 3: Thank you for your inquiry regarding the cytotoxicity assays in Panel 3g. We appreciate the opportunity to provide additional details.

You rightly pointed out that resting T cells are typically not sensitive to direct NK-mediated killing; however, there are exceptions when targeted therapies are involved. Resting T cells have been shown to be susceptible to NK-mediated ADCC when cell-surface antigen targeted therapies are employed. For example, Zanolimumab, a human CD4 monoclonal antibody, has demonstrated efficient killing of CD4+ naïve T cells through NK cell-dependent ADCC activity (Rider DA et al. Cancer Res 67: 9945-9953, 2007).

Moreover, there is strong evidence that a single dose of anti-CD52-mAb can lead to almost complete depletion of peripheral lymphocytes, including resting T cells. This can be referenced in our results in Figure 3a-c and in other publications related to anti-CD52-mAb therapies.

For the ADCC experiments in Figure 3g, we used samples from three Sézary Syndrome (SS) patients and three Mycosis Fungoides (MF) patients. These MF patients, namely MF028, MF032, and MF029, were at the late stages of their disease (IIB, IVA1, and IVB, respectively) at the time of sampling. The therapeutic antibody used in these experiments was anti-CD52-mAb.

We hope this clarifies the experimental details and the rationale behind the sensitivity of resting T cells to NK-mediated ADCC in the context of targeted therapies.

Comment 2. 5 on Figure 6: IL32 experiments. To my knowledge, IL32 is very commonly expressed in any inflammatory skin disease (and also in inflamed joints) and can be expressed by macrophages, T-cells or even fibroblasts. This is by no means specific to CTCL. The involvement of IL32 is based on correlation and mechanistic experiments (e.g. siRNA downregulation) would be needed to make a link between IL32, HLA I expression, and sensitivity to ADCC.

Thank you for your valuable suggestions. We acknowledge that IL-32 is commonly expressed in various inflammatory skin diseases, including inflammatory dermatoses, as you pointed out. To

strengthen the evidence of the specific link between IL-32 and HLA-I expression in CTCL, we conducted additional experiments.

In our new experiments, we observed that tumor T cells from MF skin lesions exhibit significantly higher expression of IL32 binding sites (IL32BC) compared to normal T cells. Furthermore, stimulation with IL32 results in the exclusive induction of classical MHC-I (HLA-A, B, C) expression on the MF tumor T cells, while normal T cells did not show this response. We have incorporated these results into Figure 6h-i, providing a clearer and more mechanistic basis for the connection between IL32 and HLA-I expression in CTCL.

We believe these findings strengthen our study's rationale and support our conclusions and have incorporated them as a new Fig. 6h-I in the revised manuscript.

POINT-BY-POINT RESPONSE TO ALL COMMENTS RAISED BY REVIEWER 3:

Reviewer #3 (Remarks to the Author): with expertise in cancer immunology, NK cell biology

In this manuscript, the authors perform ScRNA-seq on tumor cells from skin derived T cell mycosis fungoides. They find that NK cell function is down, and this may be explained by HLA-A upregulation mediated by IL-32 and that blockade (of HLA or KIR) can restore function and antitumor activity.

Comment 3.1: One of the biggest concerns I have with the manuscript is based on the NK cell biology theme. The authors interchangeably talk about MHC class I and KIR interactions but the data they have is only for HLA-A. Oddly, HLA-A, except for perhaps a few alleles, is not generally an inhibitory ligand for killer immunoglobulin-like receptors (KIR) making this story rather confusing. KIR2DL1/L2/L3 binds HLA-C (group 1 and group 2 ligands) and KIR3DL1 binds HLA-Bw4. The data in figure 1 shows HLA-A but also HLA-E, a ligand for NKG2A. Figure 2 only shows HLA-A, and the KIR ligands for HLA-B and HLA-C need to be shown to make any sense of how this is presented. Perhaps HLA-E is the dominant mechanism and monalizumab would be effective as well.

Answer 3.1: Thank you for your valuable feedback. We apologize for any confusion in our initial submission and appreciate the opportunity to clarify our NK cell biology theme.

In response to your concerns, we have taken steps to rectify the ambiguity in our manuscript. Specifically, we have added new experimental data to comprehensively address the ligands for KIR receptors. As you correctly pointed out, we initially used the abbreviation HLA-ABC, which may have created the false impression that we focus solely on HLA-A.

In reality, we have substantial evidence that, in addition to HLA-A, HLA-B and HLA-C are significantly upregulated at the protein level. We have now incorporated data from western blot analysis of MF skin T cells, clearly demonstrating this fact, as new Figures 2e and 2f in the revised manuscript. Furthermore, we have revised the text to ensure a clear and precise message.

Regarding HLA-E, we absolutely share your interest in its meanwhile proven role as an immune checkpoint. However, our experiments revealed that HLA-E expression in MF skin tumor T cells was only subtly upregulated, with considerably lower levels compared to classical MHC-I (HLA-A, B, C). This finding is presented in the newly incorporated Figure 2e, along with flow cytometry data confirming the low intensity of HLA-E expression in MF tumor cells. Even when increased in circulating tumor T cells, as observed in one CTCL patient, HLA-E expression remained markedly lower than that of HLA-ABC, as shown in the new Figure 2f.

Based on these findings, HLA-E:CD94-NKG2A interactions do not seem to play a dominant role in MF CTCL. We believe this novel discovery underscores the presence of additional MHC-I-dependent mechanism that safeguards skin T-cell tumors against NK cell eradication in mycosis fungoides, as presented in this manuscript. These insights have been thoughtfully integrated into our discussion, providing a more comprehensive understanding of our study's focus, especially in the context of Monalizumab (page 18, line 473- 483).

We hope that these additional data and clarifications effectively address your concerns and enhance the quality and clarity of our manuscript. Thank you again for your valuable input.

Comment 3.2: In figure 1g, HLA-I staining is shown. This is likely also picking up HLA-E? For extended data 4a, what is the MFI of the negative control? The correlation between HLA expression and function needs to be more clear.

Answer 3.2: Thank you for your valuable feedback on our figures. We believe this comment pertains to Fig 2g of the original submission (current Fig. 2h), not Figure 1g. We acknowledge the potential cross-reactivity of the HLA-ABC antibody clone W6/32 with HLA-E (Braud, V. Eur J Immunol. 27(5): 1164-9. 1997 (<https://doi.org/10.1002/eji.1830270517>), but we consider this less relevant for CTCL as HLA-E expression remains significantly lower than HLA-ABC in this context (see also our Response 3.1 to your Comment 3.1).

To provide a more accurate representation, we performed new experiments using flow cytometry to specifically detect HLA-E expression on tumor T cells. Our results confirm that HLA-E expression on both tumor and non-tumor T cells is indeed extremely low when compared to HLA-ABC. Furthermore, there is no significant difference observed between the levels of HLA-E expression in tumor T cells and non-tumor T cells. We have incorporated these new findings into the new Figure 2f and the new Extended Data Figure 4c.

Regarding Extended Data Figure 4d in this revised manuscript (before Extended Data Figure 4a), we have included isotype antibodies as negative controls for HLA-ABC staining. This addition ensures clarity in our methodology and allows for a better understanding of the staining specificity.

We believe these adjustments strengthen the correlation between HLA expression and function in our study. Please let us know if you have any further questions or require additional information.

Comment 3.3: The NK cells in 3F are incredibly low even in healthy skin making the physiologic relevance of the differences hard to ascertain.

Answer 3.3: We appreciate your concern regarding the low NK cell count in Figure 3F and its physiological relevance. The role of NK cells in skin T cell lymphoma lesions just starts gaining prominence in the field, with emerging data acknowledging their presence and altered phenotype. Notably, a recently published observation by Scheffschick et al. (Front Immunol. 2023 Aug 25; 14:1168684; <https://pubmed.ncbi.nlm.nih.gov/37691935/>) underscores this point.

However, we have taken your feedback, as well as that of Reviewer 2, into careful consideration. To address this issue comprehensively, we conducted additional experiments and analyses. Please also refer to our response to Comment 2.4.a for further context.

In our new experiments, illustrated in Figure 3d-e, we confirmed the presence of NK cells in both healthy individuals and typical MF skin lesions. Interestingly, the percentage of NK cells in MF patient skin is significantly higher than in healthy individuals. Moreover, our analysis of single-cell RNA datasets for various skin conditions, including MF, atopic dermatitis, basal cell carcinoma, and cutaneous B-cell lymphoma (CBCL), demonstrates that NK cells constitute approximately 2% of the immune cell population in the skin on average. Importantly, there is no significant difference in the percentage of skin NK cells between skin cancers/diseases and healthy skin biopsies, as detailed in Extended Data Figure 6 b-c.

We hope these additional findings address your concerns and contribute to a clearer understanding of the physiological relevance of NK cells in our study. If you have any further questions or require additional information, please do not hesitate to reach out.

Comment 3.4: The enhanced function with anti-KIR in figures 4e/f is modest, raising the questions above. There is confusion jumping from human to mouse. Mice do not have KIR and greater care in some of the transitions is needed. Even the abstract inaccurately talks about KIR blockade in mice in vivo as well as lines 263-4. I am not sure that the mouse data is completely relevant to the story and translatable given that mice do not have KIR. Because of all these issues, the last paragraph of the discussion seems speculative and conclusions do not seem definitive.

Answer 3.4: Thank you for your feedback regarding the mouse data and its relevance to our study. We understand your concerns and have taken steps to address them:

It's important to note that while mice do not possess KIR, they do have Ly49 receptors, which are the mouse counterparts of human KIR (Rahim, M.M. et al Front. Immuno. 5: 145. 2014

(<https://doi.org/10.3389/fimmu.2014.00145>). These receptors specifically bind to H-2Kb (MHC-I counterpart in mice) (Klein, J. Science. 203: 516-521. 1979 (<https://www.jstor.org/stable/1747152>)). In our murine study, we focused on Ly49-C/I, which are specific inhibitory receptors on NK cells in mice.

We have revised the abstract and clarified our descriptions to accurately reflect the use of Ly49 receptors in mice, ensuring that there is no confusion regarding KIR in the mouse context. We hope these clarifications address your concerns and provide a more accurate representation of our study's approach.

Regarding the Discussion part of our manuscript, we appreciate your feedback on the last paragraph's speculative nature. In response to your concern, we have reframed our discussion to provide a more comprehensive and evidence-based interpretation of our findings. We believe these changes have strengthened the clarity and reliability of our conclusions.

REVIEWER COMMENTS

Reviewer #1 (Remarks to the Author):

The authors have addressed the primary concerns of the manuscript in particular as to the difficulty in generating cell numbers from the data.

I still believe that there is one major remaining concerns:

1. I understand the constraints of the Fluidigm C1 apparatus. It still nonetheless limits the robustness of that particular finding. In light of this, I would include the data re: why or how IL-32 is dysregulated in tumor cells.

Minor concerns:

1. While I understand why the authors will present the ANN and the machine learning model to highlight their findings, it is still not clear to me that the tumor cells could not have been identified by TCR-sequencing and MHC class I not be identified by more commonly deployed single cell RNA-Seq data.

Reviewer #2 (Remarks to the Author):

I sincerely appreciate the opportunity to assess the revised manuscript. I would like to extend my gratitude to the authors for their conscientious consideration of the reviewers' comments and their earnest efforts to address the raised queries and critiques.

Upon reviewing the revised version of the paper, I must acknowledge that the narrative has been substantially refined, rendering it more coherent and reader-friendly. The strength of the evidence presented is indeed compelling. The modifications made to the figures have enhanced the overall clarity, and the additional information pertaining to methodological distinctions in single-cell sequencing techniques is a valuable addition.

In my estimation, this paper holds significant potential and will undoubtedly contribute substantially to the field. However, I would like to offer a minor suggestion regarding the

title. As it currently stands, the term "Skin T-Cell Tumors" could be interpreted as referring to macroscopic tumors typically associated with stages IIB or higher, rather than neoplastic T-cells themselves, which I believe is the authors' intended focus. To mitigate any potential ambiguity, I propose considering an alternative title, such as "MHC-I Upregulation Safeguards Neoplastic T-Cells in the Skin..." or a similar phrasing that more precisely encapsulates the paper's core theme.

Reviewer #3 (Remarks to the Author):

No further queries.

Reviewer #4 (Remarks to the Author):

Chang et al presents a manuscript where they used single-cell RNA-seq (using the Fluidigm platform) followed by TCR reconstruction with TraCeR to understand the malignant T cell clones in Mycosis fungoides, a type of cutaneous T cell lymphoma. Their analyses led to the discovery of an MHC-1-IL32 pathway that is upregulated in the malignant T cells and may negatively affect NK cell function. This was followed with functional assays that focused on NK-cell mediated ADCC and how IL32 (on malignant T cells) may modulate that process. I have some specific comments/concerns about the single-cell analysis and machine learning aspects.

1. I think the choice of the term "attention" is not adequately justified. Typically, an attention mechanism is trainable (e.g. in transformer models) and adapts during the training phase. Instead, as it is described in this paper, it looks like they are just "weights" that contributed to the final classification. This is akin to performing a 10-label classification task that returns a vector of 10 different weights (in this 40k+ labels/genes and 40k+ weights). So I don't think calling this an attention mechanism is reasonable.

2. There are many classification algorithms that can also potentially perform very well. The ANN model here resembles a linear model, so it is probably not fair to just compare its performance to other linear model e.g. logistic regression. And also considering the size of

the number of cells in the authors' dataset, which i) I do think it's fine here for performing the machine learning, and ii) is a reasonable approach for their end-goal (of identifying genes that can demarcate malignant vs non-malignant T cells), it would have been more fair to compare to other approaches such as support vector machine and tree-based classification algorithms (e.g. XGBoost), which are the state-of-the-art for explainable classification. SVM and tree-methods are well-performing and benchmarked to be superior for interpreting contributions of features. This is especially so for binary classification, like what the authors are trying to do there.

3. How does the authors' "malignant clone" definition corroborate with contemporary methods for defining malignant vs reactive T cells based on infercnv on single-cell data e.g. doi: 10.1038/s41467-022-28799-3? I quote from that paper by Liu et al. – "Malignant T cells were defined by copy number variations (CNVs) and matched TCR α and TCR β clonotypes at the single-cell level."

4. Still on the topic of the modelling I appreciate the amount of effort the authors have made in defining the clonotypes and I largely agree how they have named them and used them for the downstream analyses, although I have a slight concern about whether they are lumping reactive T cells into the "malignant" clone category (see point above 2 points). Anyway, I find the clonality diagrams very confusing, specifically the histograms. What do the percentages on the right y-axis mean?

5. In the various bar charts about the cancer related genes (1g, extended 3f and 3g), what does the x-axis actually mean? There's no proper description – it reads that the classification was performed solely based on that specific gene which doesn't sound correct? Are these supposed to be the weights (or what authors are calling attention)?

6. There's no mention in the manuscript about the number of cells acquired from the healthy donors (that the authors collected for this study) and the only description is that they were pre-enriched. Were they sorted for naïve/memory cell markers on FACS? The healthy donor single-cell data is also mostly not shown; so, from the healthy single-cell GEX data, are they naïve or memory? Would probably be better to break it down to show that all

types of expected normal T cell subtypes (naïve, central memory, effector memory, Treg, Tfh etc).

7. Pseudotime analysis – the authors claim that the trajectory followed an early-to-late disease trajectory. I don't think that's adequately justified nor generalizable as we can see that the IVA2 cells are part of a different trajectory and IVA1/IIB cells are at the end of this analysis. Trajectory analysis are not always useful and there are potentially better ways to do trajectory analysis for this kind of data structure, e.g. scdiff, which lets one build cell trajectories using a tree model where one can predefine paths based on the known branching/ordering (e.g. in this case, stages of disease). Having said that, I also don't think that the trajectory analysis adds much to the manuscript and I think can be removed.

8. The relative small number of cells due to the choice of the platform is understandable but I note that in authors' response to Reviewers 1 and 2 about this, the authors failed to comment on why wasn't the 10x 5' GEX + TCR library selected as the choice instead? Is there a logistical reason for why this is so (e.g. samples collected and process before 2019, or availability of equipment). If so, perhaps the methods section should clarify on when the samples were collected and processed?

POINT-BY-POINT RESPONSE TO THE REVIEWERS' COMMENTS:

Reviewer #1 (Remarks to the Author):

General Comment: The authors have addressed the primary concerns of the manuscript in particular as to the difficulty in generating cell numbers from the data.

General Answer: Thank you for recognizing our efforts in addressing the primary concerns, especially regarding the challenge of generating cell numbers from the data.

Comment 1.1: I still believe that there is one major remaining concerns: I understand the constraints of the Fluidigm C1 apparatus. It still nonetheless limits the robustness of that particular finding. In light of this, I would include the data re: why or how IL-32 is dysregulated in tumor cells.

Answer 1.1: Thank you for your insightful feedback. Our manuscript highlights the significant upregulation of IL-32 and increased IL-32 binding sites in tumor T cells from MF skin lesions. Further, IL-32 stimulation exclusively upregulates classical MHC-I (HLA-A, B, C) expression in MF tumor T cells. The manuscript includes a large amount of functional in vivo and in vitro data and extensively covers MHC-I upregulation in safeguarding malignant skin T-cell against NK-cell eradication in mycosis fungoides. In line with the editor's advice, we agree that exploring 'how or why IL32 is dysregulated in tumor T cells' warrants a separate follow-up research project. We appreciate your understanding.

Comment 1.2: Minor concerns: While I understand why the authors will present the ANN and the machine learning model to highlight their findings, it is still not clear to me that the tumor cells could not have been identified by TCR-sequencing and MHC class I not be identified by more commonly deployed single cell RNA-Seq data.

Answer 1.2: Thank you for your comment. Regarding the identification of tumor T cells, we utilized reconstructed TCR sequences from our single-cell RNA sequencing dataset. We also addressed your concern by applying a more commonly used analysis on single-cell RNA-Seq data [GSE173205] from 10X single-cell RNA sequencing. This method confirmed a statistically significant increase in the expression of classical MHC-I genes (HLA-A, HLA-B, and HLA-C) in tumor T cells from late-stage skin lesions. For detailed comparisons, please refer to 'Extended Data Fig. 4a' and the 'Results' section (page 8, lines 205-214).

Reviewer #2 (Remarks to the Author):

Comment 2.1: I sincerely appreciate the opportunity to assess the revised manuscript. I would like to extend my gratitude to the authors for their conscientious consideration of the reviewers' comments and their earnest efforts to address the raised queries and critiques. Upon reviewing the revised version of the paper, I must acknowledge that the narrative has been substantially refined, rendering it more coherent and reader-friendly. The strength of the evidence presented is indeed compelling. The modifications made to the figures have enhanced the overall clarity, and the additional information pertaining to methodological distinctions in single-cell sequencing techniques is a valuable addition. In my estimation, this paper holds significant potential and will undoubtedly contribute substantially to the field. However, I would like to offer a minor suggestion regarding the title. As it

currently stands, the term "Skin T-Cell Tumors" could be interpreted as referring to macroscopic tumors typically associated with stages IIB or higher, rather than neoplastic T-cells themselves, which I believe is the authors' intended focus. To mitigate any potential ambiguity, I propose considering an alternative title, such as "MHC-I Upregulation Safeguards Neoplastic T-Cells in the Skin..." or a similar phrasing that more precisely encapsulates the paper's core theme.

Answer 2.1: Thank you for your valuable comment. We appreciate your acknowledgment of our efforts. Based on your suggestion, we have modified the title to "MHC-I Upregulation Safeguards Neoplastic T Cells in the Skin Against NK Cell Eradication in Mycosis Fungoides" to enhance precision and clarity regarding our paper's focus.

Reviewer #3 (Remarks to the Author):

No further queries.

Reviewer #4 (Remarks to the Author):

Chang et al presents a manuscript where they used single-cell RNA-seq (using the Fluidigm platform) followed by TCR reconstruction with TraCeR to understand the malignant T cell clones in Mycosis fungoides, a type of cutaneous T cell lymphoma. Their analyses led to the discovery of an MHC-1-IL32 pathway that is upregulated in the malignant T cells and may negatively affect NK cell function. This was followed with functional assays that focused on NK-cell mediated ADCC and how IL32 (on malignant T cells) may modulate that process. I have some specific comments/concerns about the single-cell analysis and machine learning aspects.

Comment 4.1: I think the choice of the term "attention" is not adequately justified. Typically, an attention mechanism is trainable (e.g. in transformer models) and adapts during the training phase. Instead, as it is described in this paper, it looks like they are just "weights" that contributed to the final classification. This is akin to performing a 10-label classification task that returns a vector of 10 different weights (in this 40k+ labels/genes and 40k+ weights). So I don't think calling this an attention mechanism is reasonable.

Answer 4.1: Thank you for your valuable comment. We appreciate your insight and agree that our model can be more accurately described as a logistic regression model with "sample-specific weights". These sample-specific weights are set by a neural network. We used a hypernetwork; an attention-like adaptive mechanism, as the weights for each sample for the logistic regression model are predicted by a deep learning model.

We acknowledge the importance of clarity in terminology and have modified the wording in the manuscript accordingly. We opted to term those weights, formerly referred to as the "attention list", as "importance weights". Instead of "attention-ANN", we now refer to the model as "neural network logistic regression" or "NN-log-reg" to better convey the model's architecture.

Comment 4.2: There are many classification algorithms that can also potentially perform very well. The ANN model here resembles a linear model, so it is probably not fair to just compare its performance to other linear model e.g. logistic regression. And also considering the size of the number of cells in the authors' dataset, which i) I do think it's fine here for performing the machine learning, and ii) is a reasonable approach for their end-goal (of identifying genes that can demarcate malignant vs non-

non-malignant T cells), it would have been more fair to compare to other approaches such as support vector machine and tree-based classification algorithms (e.g. XGBoost), which are the state-of-the-art for explainable classification. SVM and tree-methods are well-performing and benchmarked to be superior for interpreting contributions of features. This is especially so for binary classification, like what the authors are trying to do there.

Answer 4.2: We thank the reviewer for highlighting other models that can potentially perform similar on the task. Although it is true that the model resembles a linear model, we want to mention that, due to the architecture, the weight for each gene is different for each cell. We agree that comparisons to Support vector machines (SVM) and Tree-based methods are needed here and we included those methods into the manuscript for comparison with our NN-log-reg model.

While testing all trained ML models, we found that our adaptive NN-log-reg model outperformed all other tested methods. The newly obtained comparisons between methods are described in the “Results” (page 7, line 163-187) and in the revised Main Figure 1e and revised Extended Data Figure 3 a-b. The related part of the Methods has been updated accordingly (page 24-25, line 649-684)

Comment 4.3: How does the authors’ “malignant clone” definition corroborate with contemporary methods for defining malignant vs reactive T cells based on infercnv on single-cell data e.g. doi: 10.1038/s41467-022-28799-3? I quote from that paper by Liu et al. – “Malignant T cells were defined by copy number variations (CNVs) and matched TCR α and TCR β clonotypes at the single-cell level.”

Answer 4.3: Thank you for your insightful comment and for drawing attention to the relevant work by Liu et al. (our Reference No. 7; doi: 10.1038/s41467-022-28799-3). Our approach to defining malignant T cells aligns with the methodology outlined in that paper. Specifically, our study defines malignant T cells by the classical criterion of TCR clonal expansion with matched TCR α and TCR β clonotypes at the single-cell level. We employed TCR reconstruction model based on TraCeR (Stubbington et al., our Reference No. 53).

As our primary focus did not involve genetic alterations, we did not include CNV analysis in the manuscript. However, to investigate potential CNV and following the same logic as in the cited study (our Reference No. 7; doi: 10.1038/s41467-022-28799-3), we performed infercnv to infer CNV from T cells that we defined as “malignant” (point-by-point figure 1).

For this analysis, we used T cells we defined as “bystanders” (equivalent to “reactive” T cells) as control (See figure below). We observed strong heterogeneity of the CNV profile of “malignant” T cells across patients. We do observe many CNVs in the “malignant” cells of some patients whereas some other patients do not seem to harbor any CNV. It is important to note that our sample size for this analysis was limited, impacting the depth of our conclusions.

*Despite this limitation, we emphasize that our approach is consistent with the Liu et al. study, where TCR reconstruction served as the initial step in distinguishing malignant from reactive T cells. Only subsequently were CNVs compared between these subsets using infercnv. This methodology is explicitly described in the Methods section of Liu et al.’s paper, reinforcing the validity of our chosen approach. (See in the Methods section of Liu et al (our Reference No. 7; doi: 10.1038/s41467-022-28799-3: “To differentiate malignant T cells from reactive T cells, large-scale CNVs were inferred from the scRNA-seq data using the R infercnv package (v1.7.1). **Reactive T cells** and other cell types **were used as controls for the CNV analysis.**”).*

Taken together, thank you again for initiating this very important discussion. We fully acknowledge the significance of genetic alterations in the context of malignant T cell definition, and while our study did not extensively delve into CNV analysis, we hope our supplementary analysis using infercnv provides valuable insights.

inferCNV

Point-by-point figure 1: Infer CNV heatmap, using T cells defined as bystanders (equivalent to reactive, as defined by TCR sequencing) as control.

Comment 4.4: Still on the topic of the modelling I appreciate the amount of effort the authors have made in defining the clonotypes and I largely agree how they have named them and used them for the downstream analyses, although I have a slight concern about whether they are lumping reactive T cells into the “malignant” clone category (see point above 2 points). Anyway, I find the clonality diagrams very confusing, specifically the histograms. What do the percentages on the right y-axis mean?

Answer 4.4: Thank you for your thoughtful comment. We appreciate your concerns about the potential lumping of reactive (“bystanders”) T cells into the “malignant” clone category. The cells, which were classified as malignant (“main-clone” or “related-to-main-clone”) were those harboring the most occurring TCR α and TCR β sequence. These malignant T-cells constituted a substantial proportion, ranging from 43% to 95%, of the total T cells within a given sample. We believe that the observed expansion in these T cells is unlikely to occur in reactive T-cell populations. Nevertheless, we do not exclude that a very limited proportion of the cells defined as “malignant” harbor the same TCR as reactive T cells. The theoretical maximum number of unique, approximately 45 bp TCR β CDR3 nucleotide sequences is 4^{45} (“Sequence analysis of T-cell repertoires in health and disease”, J. Woodsworth, Gen. Medicine 2013). Considering this, the chance of overlap between reactive and malignant T-cell clones is extremely low. Furthermore, our longitudinal data show consistent main clones across multiple years, suggesting a persistence that is less characteristic of reactive T-cell populations.

We apologize for any confusions about the histograms of TCR α -chain and β -chain. In these histograms, each column on the x-axis represents one reconstructed TCR α -chain (upper histogram) or TCR β -chain (bottom histogram) within the sample. The left y-axis indicates the actual number of cells with that specific TCR α -chain or TCR β -chain, while the right y-axis provides the percentage occurrence of each TCR α -chain or TCR β -chain within the sample. This information is now labeled to the figures and amended to the Figure Legend of Extended Data Fig 2 (Page 38, line 1020 to 1026).

Comment 4.5: In the various bar charts about the cancer related genes (1g, extended 3f and 3g), what does the x-axis actually mean? There's no proper description – it reads that the classification was performed solely based on that specific gene which doesn't sound correct? Are these supposed to be the weights (or what authors are calling attention)?

Answer 4.5: Thanks for very much for your helpful input and we apologize for not describing clearly the bar charts about the cancer related genes. As compared to regression or tree-based models, in neural network-based 'importance' methods the feature importance are different for each cell. Thus 'feature importance' is relatively difficult to visualize clearly. Therefore, we tried to find a way to display the set of most important genes for the prediction of a cell to be (A) a clonal cell or (B) a bystander cell. A gene was considered as 'important' for the prediction of a cell's class if it was among the top k (in that case top 0.5 %, this was empirically determined) genes in the weight list ('importance' values) of that specific cell. The x axis indicated the percentage of cells for which the specific gene (on the y axis) is among these top 0.5 % in the 'feature importance' list (that means 100 % for "gene A" means that "gene A" was among the top 0.5 % of 'important genes' for 100 % of correctly classified cells). Only cells that were correctly classified by the NN were considered. Attention values of genes of normalized count 0 for the specific cell were omitted. These additional explanations are now incorporated in the figure legend of Main Figure 1g and Extended Data Figure 3 d-f.

Comment 4.6: There's no mention in the manuscript about the number of cells acquired from the healthy donors (that the authors collected for this study) and the only description is that they were pre-enriched. Were they sorted for naïve/memory cell markers on FACS? The healthy donor single-cell data is also mostly not shown; so, from the healthy single-cell GEX data, are they naïve or memory? Would probably be better to break it down to show that all types of expected normal T cell subtypes (naïve, central memory, effector memory, Treg, Tfh etc).

Answer 4.6: Thank you for your valuable feedback, and we appreciate the opportunity to address your concerns. We apologize for any oversight in not explicitly mentioning the pre-enrichment method and the number of high-quality T cells obtained from healthy donors in the manuscript.

For pre-enrichment, we used SytoxRed, CD45, and CD3 fluorescent antibodies for sorting living CD45+/CD3+ total T cells via fluorescence-activated cell sorting (FACS) from both MF patients and healthy individuals. Notably, we did not employ sorting based on naïve/memory cell markers. In total, we obtained 192 high-quality T cells from 4 healthy individuals with mathematically reconstructed α or β sequences of the T-cell receptor (TCR). This information has been incorporated into the "Results" section (page 5, lines 114 and 130).

Regarding the healthy donor single-cell data, we used it as a control to illustrate the high TCR diversity in V-(D)-J combinations, compared to dominant TCR V-(D)-J combinations in MF T cells. In Figure 6 a-c, we further used the data from healthy donor single T-cell data as control to demonstrate the expression of cytokine-related genes and IL32 isoforms. We chose not to separate further the T-cell subtypes through gene signatures, also due to the limited cell number. To further differentiate T-cell subtypes, we used the expression of SELL (CD62L), CCR7, and CD44. Excluding CD8-expressing T cells, we observed that 44.7% were naïve T cells (CD62L+ or CCR7+, and CD44-), 29.4% were central memory T cells (CD62L+ or CCR7+, and CD44+), 21.2% were effector memory T cells (CD62L-, CCR7-, and CD44+),

and 4.7% were classified as other T cells (Treg, Tfh, etc.). These details, along with feature plots of the expression of relevant markers on healthy T cells, have been included in Point-by-point figure 2.

Point-by-point figure 2: The feature plots of the expression of CD3E, PTPRC (CD45), SELL (CD62L), CCR7 and CD44 on healthy T cells.

Comment 4.7: Pseudotime analysis – the authors claim that the trajectory followed an early-to-late disease trajectory. I don't think that's adequately justified nor generalizable as we can see that the IVA2 cells are part of a different trajectory and IVA1/IIB cells are at the end of this analysis. Trajectory analysis are not always useful and there are potentially better ways to do trajectory analysis for this kind of data structure, e.g. scdiff, which lets one build cell trajectories using a tree model where one can predefine paths based on the known branching/ordering (e.g. in this case, stages of disease). Having said that, I also don't think that the trajectory analysis adds much to the manuscript, and I think can be removed.

Answer 4.7: Thank you for your valuable input. We fully agree and appreciate your comment regarding the pseudotime/trajectory analysis, which we removed from the manuscript. Instead, we have incorporated as new Figure 2b SuperPlots illustrating the gene expression of the top important cancer-related genes identified via the NN-log-reg method to be associated with the real-time disease stage that patients reached either at death or within 5-year follow-up period. The revised approach with SuperPlots effectively conveys the relevant information, specifically highlighting the significant escalation in the expression of MHC-I, IFITM1, and IFITM2 as the MF progresses. The updated content can be found in the "Results" section (page 8, lines 199 and 204) and the "Figure Legends" (page 34, lines 899 and 902).

Comment 4.8: The relative small number of cells due to the choice of the platform is understandable but I note that in authors' response to Reviewers 1 and 2 about this, the authors failed to comment on why wasn't the 10x 5' GEX + TCR library selected as the choice instead? Is there a logistical reason

for why this is so (e.g. samples collected and process before 2019, or availability of equipment). If so, perhaps the methods section should clarify on when the samples were collected and processed?

Answer 4.8: We appreciate your understanding of the relatively small number of cells, and we acknowledge the importance of providing clarity on the platform selection. Indeed, the decision to use the Fluidigm C1 platform was influenced by availability of the equipment and the timeline of sample collection and processing, which occurred between 2013 and 2017 for sample collection and between 2016 and 2018 for processing. The time between 2019 and 2022 was dedicated to the functional in vitro and especially in vivo experiments, presented in Figures 2-6. Following your advice, we have incorporated information about the sample collection and processing timeline in the "Methods" section (page 23, lines 605 and 606). We hope this addition provides the necessary context for the choice of platform and addresses your concerns.

REVIEWERS' COMMENTS

Reviewer #4 (Remarks to the Author):

I thank the authors for carefully addressing my previous comments and I am largely satisfied with the responses. I have few minor comments below for the authors to fix up.

Congratulations on a great manuscript!

1) Thank you for addressing my main point about the use of the term "attention".

However, I note that there are still a few instances where "attention" is being used in the wording which I would like to authors to change to using "importance weight" or relevant term:

Fig 1d

Fig 6A

Page 25, line 671

github repository readme <https://github.com/suskim/att-ANN>

2) I also note that the last update for the single-cell RNA seq analysis scripts at <https://github.com/GuenovaLab/TumorLymphocytes> was ~9 months ago. Can the authors look into this and update the repo with relevant revised scripts (especially the code for generating for the new superplots)?

3) On the topic of the superplots - thank you for this. The results from this is definitely more compelling. However, the legend is very brief. Clarification of the error bars and larger solid points would be useful? I'm guessing they are the mean values of the sample and the error bars are the SEM of these mean values?. Were the statistical tests done on the sample level or on the data from the individual cells? Are the values log/raw counts?

4) Thank you for attempting to address the CNV analysis. I think the authors should include a version of the response in the discussion as a "limitation" of the study - the readers should know that you have attempted to address this and while the CNV results showed some trends, there may still be a chance (however unlikely it may be statistically) be some reactive cells that are grouped into the malignant cells.

5) Fig 6c has a typo - "exrepssion"

Reviewer #4 (Remarks on code availability):

1) Still says it's an attention neural network in <https://github.com/suskim/att-ANN>

2) the python code looks usable enough although there's no example data, no CI/CD testing, and no unittests to verify whether the packages can be installed/reproduced.

3) The last update for the single-cell RNA seq analysis scripts at <https://github.com/GuenovaLab/TumorLymphocytes> was ~9 months ago. Can the authors look into this and update the repo with relevant revised scripts (especially the code for generating for the new superplots)?

POINT-BY-POINT RESPONSE TO THE REVIEWERS' COMMENTS:

Reviewer #4 (Remarks to the Author):

Overall Comment: I thank the authors for carefully addressing my previous comments and I am largely satisfied with the responses. I have few minor comments below for the authors to fix up. Congratulations on a great manuscript!

Answer: Thank you for your positive feedback and thoughtful comments. We appreciate your support and have addressed all minor points you've raised.

Comment 1: Thank you for addressing my main point about the use of the term "attention". However, I note that there are still a few instances where "attention" is being used in the wording which i would like to authors to change to using "importance weight" or relevant term:

Fig 1d

Fig 6A

Page 25, line 671

github repository readme <https://github.com/suskim/att-ANN>

Answer 1: Thank you very much for noticing this. We modified the "attention" to "importance weight" on Fig 1d, page 25 line 671 and we changed the "attention-ANN" or "att-ANN" to "NN-log-reg" on Fig 6a and the github repository readme. The new link of github is <https://github.com/suskim/NN-log-reg>.

Comment 2: I also note that the last update for the single-cell RNA seq analysis scripts at <https://github.com/GuenovaLab/TumorLymphocytes> was ~9 months ago. Can the authors look into this and update the repo with relevant revised scripts (especially the code for generating for the new superplots)?

Answer 2: Thank you. We added the SuperPlot scripts, updated all scripts related to scRNA analysis and public analysis, and cleaned the repository of unused codes.

Comment 3: On the topic of the superplots - thank you for this. The results from this is definitely more compelling. However, the legend is very brief. Clarification of the error bars and larger solid points would be useful? I'm guessing they are the mean values of the sample and the error bars are the SEM of these mean values?. Were the statistical tests done on the sample level or on the data from the individual cells? Are the values log/raw counts?

Answer 3: Indeed, we agree that the description was too scanty. We rectified the legend of figure 2b: Each dot represents normalized expression of the given marker in a single cell. The larger solid points represent mean normalized expression per patient sample. The black bars represent mean and standard error of the mean (SEM) of the larger solid points in each stage. The two-sided student t-test was used to compare normalized expression at the single-cell level taking the early stage (I-IIA) as reference. For HLA-A and HLA-C, the normalized expression of all transcripts was summed [Legends; Page 38, Figure 2b].

Comment 4: Thank you for attempting to address the CNV analysis. I think the authors should include a version of the response in the discussion as a "limitation" of the study - the readers should know that you have attempted to address this and while the CNV results showed some trends, there may still be a chance (however unlikely it may be statistically) be some reactive cells that are grouped into the malignant cells.

Answer 4: Thank you for your valuable suggestion. We included a sentence in the discussion describing that there may still be a small chance to group bystander T cells as malignant if they share the same TCR as malignant T cells [Discussion, Page 16, Line 431-432].

Comment 5: Fig 6c has a typo - "exrepsion"

Answer 5: Thank you for noticing the typo on Fig6c. We corrected it.

Reviewer #4 (Remarks on code availability):

1) Still says it's an attention neural network in <https://github.com/suskim/att-ANN>

Answer: Thank you for bringing this to our attention. We have updated the GitHub repository to reflect the correct information. The new link is <https://github.com/suskim/NN-log-req>

2) the python code looks usable enough although there's no example data, no CI/CD testing, and no unittests to verify whether the packages can be installed/reproduced.

Answer: We appreciate your feedback and have included example data for verification.

3) The last update for the single-cell RNA seq analysis scripts at

<https://github.com/GuenovaLab/TumorLymphocytes> was ~9 months ago. Can the authors look into this and update the repo with relevant revised scripts (especially the code for generating for the new superplots)?

Answer: Thank you. In response to your suggestion, we have added SuperPlot scripts, updated all scripts related to scRNA analysis and public analysis, and cleaned the repository of unused codes.